# Advancing Prompt-Based Methods for Replay-Independent General Continual Learning

**Zhiqi Kang**
Inria*

**Liyuan Wang**[†]
Tsinghua University

**Xingxing Zhang**
Tsinghua University

**Karteek Alahari**[†]
Inria*

## Abstract

General continual learning (GCL) is a broad concept to describe real-world continual learning (CL) problems, which are often characterized by online data streams without distinct transitions between tasks, i.e., blurry task boundaries. Such requirements result in poor initial performance, limited generalizability, and severe catastrophic forgetting, heavily impacting the effectiveness of mainstream GCL models trained from scratch. While the use of a frozen pretrained backbone with appropriate prompt tuning can partially address these challenges, such prompt-based methods remain suboptimal for CL of remaining tunable parameters on the fly. In this regard, we propose an innovative approach named MISA (Mask and Initial Session Adaption) to advance prompt-based methods in GCL. It includes a forgetting-aware initial session adaption that employs pretraining data to initialize prompt parameters and improve generalizability, as well as a non-parametric logit mask of the output layers to mitigate catastrophic forgetting. Empirical results demonstrate substantial performance gains of our approach compared to recent competitors, especially without a replay buffer (e.g., up to 18.39, 22.06, and 11.96 % points performance lead on CIFAR-100, Tiny-ImageNet, and ImageNet-R, respectively). Moreover, our approach features the plug-in nature for prompt-based methods, independence of replay, ease of implementation, and avoidance of CL-relevant hyperparameters, serving as a strong baseline for GCL research. Our source code is publicly available at https://github.com/kangzhiq/MISA.

## 1 Introduction

Continual learning (CL) (Wang et al., 2024b; De Lange et al., 2021) focuses on the lifelong acquisition of knowledge in response to real-world changes. Although conventional CL research often relies on offline training of each task with distinct transitions between tasks (i.e., clear task boundaries), online scenarios with blurry task boundaries tend to be more practical yet challenging. Such "online" scenarios demand the model to swiftly adapt to new information, which is essential for real-time applications. Blurry task boundaries further reflect realistic data distributions without distinct transitions between tasks, i.e., old classes disappear gradually over time and might re-appear when new classes emerge (see Fig. 1, Left). The above considerations, collectively referred to as general continual learning (GCL) (De Lange et al., 2021), have received increasing attention in recent years. However, most of the earlier works (Aljundi et al., 2019; Buzzega et al., 2020; Koh et al., 2021; Bang et al., 2021) address GCL by learning a model from scratch. This strategy often leads to challenges such as poor initial performance, limited generalizability, and severe catastrophic forgetting (McCloskey & Cohen, 1989) in such a complex setting.

One promising direction that emerged in GCL literature (Moon et al., 2023) is to employ prompt-based methods (Wang et al., 2022d;c; 2024a) on the basis of pretrained models (PTMs), which involves keeping the PTMs frozen and introducing a few prompt parameters for representation learning. Such methods outperform previous train-from-scratch ones by a significant margin. The frozen backbone pretrained on a large-scale dataset provides a solid initialization with strong generalizability and is resistant to forgetting. However, the particular challenges of GCL persist for the remaining

---

[†]Corresponding authors: L. Wang and K. Alahari.
*Univ. Grenoble Alpes, CNRS, Grenoble INP, LJK, 38000 Grenoble, France.

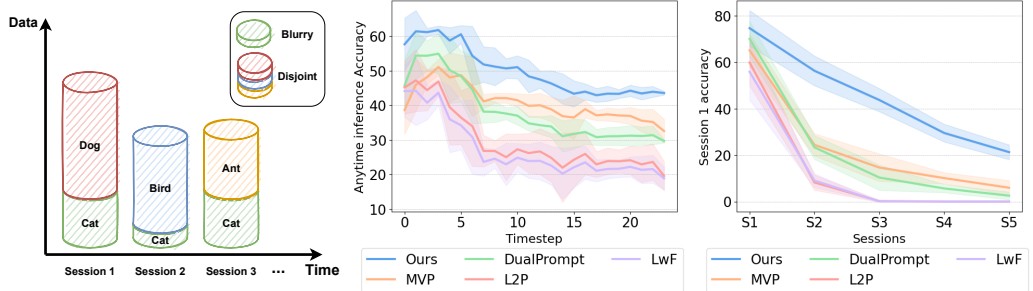

Figure 1: Problem setup and motivation. Left: illustration of the GCL data stream. Mid: average prediction accuracy at different timesteps in GCL. Right: session 1 accuracy, where we evaluate the retention of knowledge acquired at session 1 after each session. All methods are tested without a replay buffer.

tunable components, i.e., the prompt parameters and the output layers, making the direct application of prompt-based methods to GCL suboptimal.

Studies have demonstrated that prompt parameters and output layers are particularly vulnerable in the challenging setting of GCL. First, prompt-tuning is known to suffer from a limited capacity in vision tasks (Lester et al., 2021; Vu et al., 2021; Chen et al., 2021). Training these parameters on the fly in online scenarios of GCL can result in poor initial performance and limited generalizability. As shown in Fig. 1 (Mid), state-of-the-art methods have moderate initial performance and drop dramatically after session transitions (timestep 4). Moreover, it is well studied in the CL literature (Wu et al., 2019; Ahn et al., 2021) that the output layers suffer from forgetting in class-incremental learning, a challenge that is exacerbated in GCL due to the naturally imbalanced data stream and the interference between new and old knowledge caused by blurry task boundaries. While recent GCL works (Moon et al., 2023) have introduced specialized learning strategies for the output layers, they still fail to outperform conventional prompt-based methods. Specifically, the session 1 accuracy in Fig. 1 (Right) shows the capacity to retain acquired knowledge from the first session in subsequent sessions, where all existing methods suffer to retain this knowledge. Thus, there is a clear need for learning-efficient and forgetting-less strategies to address the particular challenges of GCL.

Our method, MISA (Mask and Initial Session Adaption), is designed with two objectives: a forgetting-aware initial session adaption (ISA) for better learning efficiency and generalizability of the prompt parameters, and a non-parametric logit masking for less catastrophic forgetting of the output layers. The initial session refers to a warm-up of the model parameters prior to any GCL sessions. Specifically, we reuse the pretraining data of the PTMs to warm up the prompt parameters. As naïve pretraining tends to enhance mainly the initialization rather than the generalizability, we devise a novel forgetting-aware minimization technique in ISA to improve the generalizability of prompts to distribution shift. Unlike conventional CL methods that address forgetting after it occurs, our approach proactively prevents future forgetting and maintains the pretrained knowledge for downstream GCL. As for reducing forgetting of the output layers, we propose a non-parametric strategy that is based on class appearance to mask the output logits. We show that such a simple strategy, which is not effective for the train-from-scratch paradigm, can benefit from the stable representation space retained by the frozen backbone and effectively rectify the output layers.

In summary, our MISA is specifically designed for tunable parameters of prompt-based methods in GCL. We emphasize the plug-in nature of MISA for different methods and scenarios, as it is replay-independent, hyperparameter-free, and effective in GCL. Accordingly, our approach brings substantial performance gains compared to recent strong baselines. For instance, there is up to 18.39, 22.06, and 11.96 % points performance lead on CIFAR-100, Tiny-ImageNet, and ImageNet-R datasets, respectively. Our contributions can be summarized as follows:

- We propose an innovative approach that addresses the poor initialization, limited generalizability, and severe forgetting of GCL, which features the ease of implementation, avoidance of CL-relevant hyperparameters, and plug-in nature for prompt-based methods.

- Our proposed forgetting-aware initial session adaptation effectively improves the initialization of prompt parameters and their generalizability to distribution shift, and our non-parametric logit masking rectifies the output layers to alleviate catastrophic forgetting.

- Across multiple challenging benchmarks, our approach outperforms a variety of existing methods by a wide margin, setting a new state of the art for GCL research.

## 2 RELATED WORK

**Continual Learning (CL).** The default setting of CL involves offline training of each task with explicit task boundaries. Representative methods for this setting can be divided into three groups (Wang et al., 2024b). The first one is *replay-based methods*, which store or generate a few old training samples for subsequent reuse (Buzzega et al., 2020; Wang et al., 2022a; Kang et al., 2023; Ostapenko et al., 2019). The second is *parameter isolation methods* (Serra et al., 2018; Jung et al., 2020), which allocates task-specific parameters to prevent overwriting. The third is *regularization-based methods*, which introduce regularization in loss functions to mitigate forgetting (Wu et al., 2019; Cha et al., 2021; Douillard et al., 2020). There have been attempts to regularize the sharpness of the loss surface to alleviate forgetting (Mirzadeh et al., 2020; Yang et al., 2023; Mehta et al., 2023), but these are limited to a training-from-scratch paradigm or shallow networks.

Prompt-based methods (Smith et al., 2023; Wang et al., 2022d;c; 2024a) provide another avenue for CL. In particular, they are state of the art for conventional CL settings, wherein the PTMs are frozen and a few prompt parameters are introduced for representation learning. They can be categorized into task-specific prompts (Razdaibiedina et al., 2023a; Wang et al., 2022b), shared-task prompts (Wang et al., 2022d; Smith et al., 2023) and their variants (Wang et al., 2022c; Hong et al., 2024). Although these methods perform well in conventional CL scenarios with adequate pretraining, they exhibit many problems in GCL, such as poor initial performance, limited generalizability, and severe catastrophic forgetting. Moreover, many advanced prompt-based methods are not adapted to GCL by design. For instance, CODA-Prompt (Smith et al., 2023) and NSP$^2$ (Lu et al., 2024) assume orthogonality between tasks, which cannot be satisfied in the case of blurry task boundaries. RanPAC (McDonnell et al., 2024) and HiDe-Prompt (Wang et al., 2024a) lack the design for online updates of class statistics in GCL.

**General Continual Learning (GCL)** is a broad concept to describe a variety of practical challenges in CL (De Lange et al., 2021). In addition to the primary focus on "online learning" and "blurry task boundaries" considered in our approach, other requirements of GCL like "constant memory" and "no test-time oracle" also play a crucial role in enabling the model to better adapt to real-world applications. These additional aspects will be briefly discussed in this paper as well to demonstrate that our approach is suitable for GCL. Various settings have been proposed as realizations of GCL (Aljundi et al., 2019; Koh et al., 2021; Moon et al., 2023). Si-Blurry (Moon et al., 2023) is one of the recent approaches for GCL to evaluate the two primary facets, which assumes that the distributions of training samples belonging to each class are randomly sampled in each task. Although there have been numerous attempts (Aljundi et al., 2019; Buzzega et al., 2020; Bang et al., 2021) to address GCL, they often focus on training from scratch and thus require replaying a few old training samples, achieving inferior performance with potential privacy issues. Gummadi et al. (2022) dealt with blurry task boundaries with novelty detection yet is designed for offline scenarios. The most recent and only prompt-based GCL method (Moon et al., 2023) demonstrated improved performance when using a pretrained backbone compared to training from scratch. However, it remains suboptimal in addressing the GCL challenges, yielding only marginal improvements over conventional prompt-based CL methods.

## 3 PRELIMINARIES

In this section, we review the problem setup of conventional CL and GCL, as well as the advanced prompt-based methods that address the former.

**Problem Setup.** Let $\mathbb{C}$ be the set of available classes whose cardinality is denoted as $N = |\mathbb{C}|$. Suppose that there are $T$ learning sessions or tasks. In conventional CL, we create a uniform partition $\{\mathbb{C}_t\}_{t=1...T}$ of $\mathbb{C}$, where $\mathbb{C}_t$ represents the classes assigned to task t. The number of classes in each task is identical, i.e., $\forall i, j, i \neq j, |\mathbb{C}_i| = |\mathbb{C}_j|$. In addition, at each task, the model can extensively iterate on the data with epoch $K \in \mathbb{N}$. In contrast, GCL relaxes the assumptions of clear boundaries and offline learning. Due to such blurry task boundaries, we opt for the notion of learning sessions,

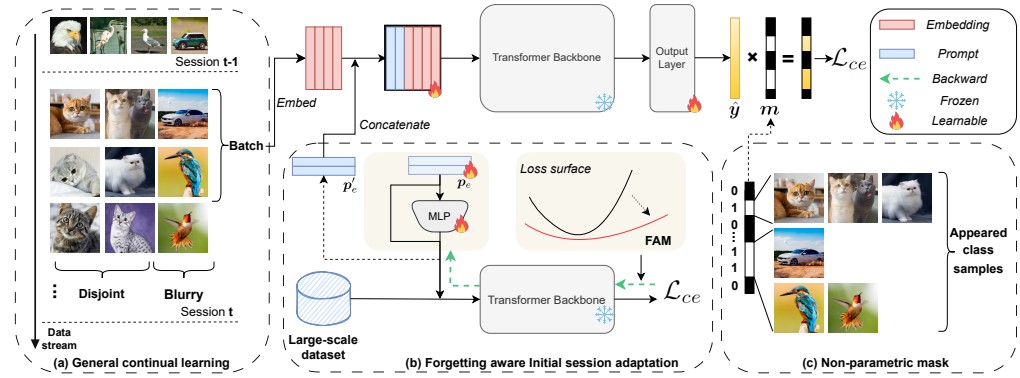

Figure 2: An overview of our MISA with a frozen pretrained backbone in GCL. (a) Data in GCL consists of disjoint and blurry classes. (b) Initial session adaption is conducted prior to any CL sessions. Once finished, only the warmed-up prompt parameters are reused for CL. (c) Non-parametric logit mask which retains logits of available classes in a batch or a session.

which represent different time steps in a one-pass data stream of GCL. Specifically, classes can overlap between learning sessions, such that:

$$\forall i, j, i \neq j, P(\mathbb{C}_i \cap \mathbb{C}_j \neq \emptyset) > 0, \tag{1}$$

where $P$ denotes probability. Moreover, due to the one-pass constraint on the data, the model can only perform one epoch over the data, such that $K = 1$. Consequently, GCL emphasizes realistic data distributions and the swift adaptation of the model.

The latest realization of GCL is Si-Blurry (Moon et al., 2023). Here, the available classes $\mathbb{C}$ are randomly divided into non-overlapping $\mathbb{C}^D$ and $\mathbb{C}^B$ for disjoint classes and blurry classes, respectively. The *disjoint class ratio* is defined as $m = |\mathbb{C}^D|/|\mathbb{C}|$. $\mathbb{C}^D$ and $\mathbb{C}^B$ are divided into non-uniform partitions $\{\mathbb{C}^D_t\}_{t=1...T}$ and $\{\mathbb{C}^B_t\}_{t=1...T}$, unlike the uniform distribution in conventional CL. Although all data corresponding to $\mathbb{C}^D$ is divided into disjoint sessions, there is a ratio of $n$ training samples from $\mathbb{C}^B$ being randomly shuffled across sessions, with $n$ as the *blurry sample ratio*. The stochastic nature of Si-Blurry is controlled by $m$ and $n$. As a realization of GCL, the shuffling of blurry classes satisfies Eq. 1. Moreover, Si-Blurry assumes an online scenario that naturally fits in the one-pass requirement of GCL. We also show in Appendix A.3 that Si-Blurry is a realization of generalized class incremental learning (Mi et al., 2020).

**Prompt-Based Methods.** We denote a pretrained vision transformer (ViT) (Dosovitskiy et al., 2020) as $f = f_c \circ f_r \circ f_e$, where $f_e$ is the input embedding network, $f_r$ is a stack of self-attention layers, and $f_c$ is the output layer(s). For an input image $\boldsymbol{x}$ and its one-hot label vector $\boldsymbol{y}$, let $f_e(\boldsymbol{x}) = \boldsymbol{x}_e \in \mathbb{R}^{L \times D}$ be the embedding features with $L$ the token length and $D$ the embedding dimension. Prompts are learnable parameters $\boldsymbol{p}_e \in \mathbb{R}^{L_p \times D}$ prepended to the embedding features as $\boldsymbol{x}_p = [\boldsymbol{p}_e; \boldsymbol{x}_e]$, with $L_p$ the prompt length. The extended features are forwarded to the network such that $\hat{\boldsymbol{y}} = f_c \circ f_r(\boldsymbol{x}_p)$, with $\hat{\boldsymbol{y}} \in \mathbb{R}^N$ the prediction vector. Existing prompt-based methods (Wang et al., 2022d;c; Smith et al., 2023) for conventional CL usually train a pool of prompts and at each time select the most relevant ones through a query-key matching mechanism. Consequently, the poorly trained prompts can not only impair representation learning but also disrupt the matching mechanism, leading to degraded performance. Therefore, enhancing the knowledge encoded in prompts is crucial for improving these methods.

## 4 METHOD

We now present the two main components of our MISA, as shown in Fig. 2: (1) forgetting-aware initial session adaption (ISA), which warms up prompt parameters with forgetting-aware minimization (FAM) and prompt augmentation; and (2) non-parametric logit masking that rectifies the output of the model to avoid the interference of old and new knowledge. These two components provide benefits such as ease of implementation, avoidance of CL-relevant hyperparameters, and their plug-in nature for other prompt-based methods.

## 4.1 INITIAL SESSION ADAPTION

We start by defining the *initial session*. In conventional (offline) CL, it has been overlooked that using the pretrained models for prompt-based methods implies an initial learning session of the backbone prior to any downstream CL sessions. To fully exploit the benefits of pretraining data, we propose to incorporate prompt parameters in the initial session of GCL. In particular, we leverage the pretraining dataset $\mathcal{D}_{init}$ to prepare the prompt parameters for swift adaption. We warm up the prompt parameters in a supervised manner[*] for ISA, following the pretraining paradigm of the backbone. Accordingly, the supervised learning objective is defined as:

$$\min_{\boldsymbol{p}_e, f_c} \sum_{(\boldsymbol{x}, \boldsymbol{y}) \in \mathcal{D}_{init}} \mathcal{L}_{ce}(f_c \circ f_r([\boldsymbol{p}_e; \boldsymbol{x}_e]), \boldsymbol{y}), \tag{2}$$

where $\mathcal{L}_{ce}$ is the cross-entropy loss. We omit the regularization term in Eq. 2 for simplicity. Both $\boldsymbol{p}_e$ and $f_c$ are randomly initialized. The learned $f_c$ will be discarded after this step, and only the learned $\boldsymbol{p}_e$ will be reused afterwards. The class prototypes learned from $\mathcal{D}_{init}$ in $f_c$ are irrelevant to downstream tasks, as class identities are expected to change, making them ineffective for downstream use. Note that using the same pretraining dataset at the backbone is a design choice to ensure a fair comparison with existing methods, as the adaptation of $\boldsymbol{p}_e$ does not require any additional data, rather than a strict requirement of the ISA dataset. Additional experiments when pretraining data is partially or completely unavailable can be found in Appendix A.4.9.

There are several reasons for which ISA should naturally be effective. First, ISA for the prompt parameters is equivalent to the pretraining operation in an offline setting, which has already been shown effective in various settings. Second, ISA can also be seen as a particular initialization of prompts, which proved to be essential for the effectiveness of visual prompt tuning (Tsai et al., 2024; Shen et al., 2024). Lastly, from a domain adaptation perspective, the warmed-up prompts effectively reduce the domain gap to downstream GCL compared to randomly initialized prompts. It should be noted that ISA is designed to improve the poor learning capacity of visual prompts in GCL. Such limitation is less significant in conventional (offline) CL, as sufficient training with adequate differentiation of task-specific knowledge allows the prompts to eventually converge to a near-optimal stage. This highlights that our design is tailored to the particular challenges of GCL.

## 4.2 FORGETTING-AWARE MINIMIZATION

While naïvely initializing the prompt parameters using Eq. 2 on pretraining data can address issues due to poor initialization, this straightforward strategy lacks considerations for generalizability. Inspired by sharpness-aware minimization (SAM) that encourages the model to converge to a flat minima, we propose forgetting-aware minimization (FAM) in ISA, which seeks forgetting-aware flat minima and improves generalizability to distribution shift of prompt parameters.

**Vanilla SAM.** SAM (Foret et al., 2020) aims to reduce the sharpness of the loss surface by a minimax game: minimizing the maximal loss changes from a small perturbation $\epsilon$ on the model's learnable parameters $\theta$. The optimization objective is defined as:

$$\min_{\theta} \max_{||\epsilon||_2 \leq \rho} \mathcal{L}_{\text{train}}(\theta + \epsilon), \tag{3}$$

where $\rho$ is the size of the neighborhood and $\mathcal{L}_{\text{train}}$ represents a general learning objective. The correlation between the flatness of the loss surface and less forgetting in CL has been revealed by Mirzadeh et al. (2020); Shi et al. (2021); Mehta et al. (2023), which motivates the use of SAM in CL (Yang et al., 2023). However, previous efforts mainly focus on a training-from-scratch paradigm of shallow ResNets, rather than a pretraining paradigm of large transformer backbones.

**Perturbation in Vanilla SAM.** The optimal perturbation $\epsilon^\star$ can be calculated by using a first-order Taylor expansion of $\mathcal{L}_{\text{train}}(\theta + \epsilon)$ around $\theta$. This simplifies the original optimization problem into a linear-constrained one, which can be approximated as (Foret et al., 2020):

$$\epsilon^\star \overset{\Delta}{=} \arg\max_{||\epsilon||_2 \leq \rho} \mathcal{L}_{\text{train}}(\theta + \epsilon) \approx \rho \frac{\nabla_\theta \mathcal{L}_{\text{train}}(\theta)}{\|\nabla_\theta L_{\text{train}}(\theta)\|_2}. \tag{4}$$

---

[*]We leave self-supervised warm-up as a potential future direction.

Although this strategy has shown effectiveness in various situations, applying SAM directly to CL neglects the inherent conflict caused by distribution shift – a common challenge in CL. In particular, when the distribution of a new task diverges significantly from that of the previous task(s), the model is likely to be pulled from the initial flat loss surface to a sharp and unconstrained one. This phenomenon is widely recognized as a cause of forgetting, but here, it also contributes to a loss of generalizability of the initialized prompt parameters. Therefore, the perturbation in vanilla SAM is not a reliable estimation of future forgetting in CL, significantly limiting its effectiveness. To address this issue, our approach is to simulate such distribution shifts during ISA and estimate perturbations from them, thereby enhancing the robustness of the initialized parameters against forgetting and the loss of generalizability.

**Forgetting-Aware Minimization.** Our design improves generalizability by using gradients computed for external tasks as perturbations. We define the data from the target task as in-distribution data $\mathcal{D}_{id}$ and the data from the external task as out-of-distribution data $\mathcal{D}_{ood}$. While also minimizing the training loss under parameter perturbations, we target the perturbations that are consistent with the external task gradient, rather than those that lead to the maximum changes of the training loss. Consequently, this gradient represents the maximum forgetting for $\mathcal{D}_{id}$ data caused by $\mathcal{D}_{ood}$. Such perturbation pulls the model out of the loss space of $\mathcal{D}_{id}$ and encourages the model to flatten as well the loss surface for out-of-distribution data. This is a realistic simulation of the parameter update for downstream CL tasks after ISA. The overall optimization objective is defined as:

$$\min_{\theta} \quad \mathcal{L}_{\text{train}}(\theta + \delta; \mathcal{D}_{id}), \tag{5}$$

$$\text{subject to} \quad \delta = -\rho \frac{\nabla_{\theta} \mathcal{L}_{\text{train}}(\theta; \mathcal{D}_{ood})}{\|\nabla_{\theta} L_{\text{train}}(\theta; \mathcal{D}_{ood})\|_2}. \tag{6}$$

Similar to SAM, $\rho$ is a constant controlling the neighborhood radius. Like SAM, our design requires only one additional backpropagation step. In practice, to avoid any additional data for ISA, we simply split the ISA dataset $\mathcal{D}_{init}$ without overlap into a large subset of $\mathcal{D}_{id}$ and a small subset of $\mathcal{D}_{ood}{}^{*}$.

In summary, our FAM aims to not only find a flat loss surface, but also mitigate the loss of generalizability and performance for parameters initialized from ISA by proactively preventing future forgetting. Unlike conventional CL methods that address forgetting after it occurs, we take a preventative approach with the help of pretraining. While vanilla SAM offers some benefits in this regard, its perturbation is randomly chosen from the high-dimensional loss space unrelated to forgetting, and is ineffective in handling distribution shifts in CL. Accordingly, we use ISA data to create pseudo-downstream tasks, optimizing on a more targeted and informative direction of perturbation to obtain a more forget-aware flat loss surface. Notably, Eq. 5 and Eq. 6 exhibit a similar form to MAML (Finn et al., 2017) (i.e., a representative meta-learning framework), suggesting that our FAM promotes generalizability of forgetting-sensitive directions in a data-driven manner.

**Prompt Augmentation.** In preliminary experiments, we observe that vanilla SAM faces difficulties in optimizing in the limited and constrained space of the prompt parameters (see Appendix A.4.3). Therefore, we propose to use a prompt augmentation technique in ISA to directly increase the complexity of the learnable parameter space. Specifically, we have $\boldsymbol{p}'_e = \boldsymbol{p}_e + f_{\text{MLP}}(\boldsymbol{p}_e)$ with $f_{\text{MLP}}$ is a shallow multi-layer perception (MLP) (Razdaibiedina et al., 2023b;a). With the additional parameters and non-linearity introduced by the MLP, the learning capacity is improved to enable the flatness-aware minimization for prompt-based methods. At the end of ISA, $f_{\text{MLP}}$ will be discarded and we store the augmented $\boldsymbol{p}'_e$ for later purposes, as shown in Fig. 2 (b).

### 4.3 Non-Parametric Logit Masking

**Logit Masking.** Although the blurry task boundaries of GCL result in a natural replay that partially alleviates forgetting, they make it more difficult for the model to handle the balance between learning and forgetting, especially in the output layers. Specifically, GCL not only leads to an imbalance in class distribution but also enforces the model to overfit the re-appeared classes $\mathbb{C}^B$ and thus

---

$^{*}$We leave the choice of external datasets as a future direction since the focus of this work is to enhance prompt parameters without demanding any external data.

overwrite the knowledge of other previous tasks, especially the disjoint classes $\mathbb{C}^D$. This is a typical cause of forgetting when the model is trained with a supervised cross-entropy loss $\mathcal{L}_{ce}$. Output logit masking is known to be effective for this in conventional CL, as it isolates the logits of non-repeated classes to maintain their activation at test time. We have:

$$\min_\theta \sum_{(\boldsymbol{x}, \boldsymbol{y}) \in \mathcal{D}} \mathcal{L}_{ce}(\boldsymbol{m} \odot \hat{\boldsymbol{y}}, \boldsymbol{y}), \tag{7}$$

where $\mathcal{D}$ is the dataset available and $\odot$ is an element-wise multiplication, $\boldsymbol{m} \in \mathbb{R}^N$ is a masking vector that masks out some logits and keeps the rest. One of the latest works in GCL, MVP (Moon et al., 2023), trains a learnable mask to improve information retention. We empirically find that such a learnable mask is not desirable, as the method still suffers from strong forgetting to retain the knowledge of previous classes (see Fig. 1, Right).

**Non-Parametric Logit Masking.** After analyzing the purpose of the logit mask and the sub-optimality of existing learnable masks, we propose a simple logit mask that is parameter-free and can effectively reduce the information interference between classes. The first masking strategy is a session-level mask, where the mask is renewed at the end of each session and keeps the logits of available classes in the current session. However, it requires the notion of sessions, which is not always meaningful in an online setting. To this end, we further propose a batch-level mask that operates in the same way but at the batch level. We thus have:

$$\boldsymbol{m}_i = \begin{cases} 1, & \text{if } \boldsymbol{y}_i = 1, \text{for } (\boldsymbol{x}, \boldsymbol{y}) \in \mathcal{D}, \\ 0, & \text{otherwise}, \end{cases} \tag{8}$$

where $\boldsymbol{y}_i$ is the $i$-th entry of the one-hot label vector, and $\mathcal{D}$ is the set of available data. More precisely, when $\mathcal{D}$ refers to the data from a mini-batch, $\boldsymbol{m}$ is a batch-level mask, as shown in Fig. 2 (c). Instead, when $\mathcal{D}$ refers to the data available for one session, $\boldsymbol{m}$ is a session-level mask. We then multiply the mask with the prediction vector as defined in Eq. 7.

Despite its simplicity, non-parametric masking brings a substantial improvement in reducing forgetting, which allows the model to better adapt to GCL. The effectiveness of this simple logit masking arises from the disentanglement of representation learning and learning the classifier. With a frozen PTM ensuring a stable representation space, the mask effectively mitigates information interference in the output layers. In contrast, training-from-scratch approaches update all parameters simultaneously, reducing the efficacy of the logit mask. We include further analysis and comparison with the existing logit masking strategy Caccia et al. (2021) in Appendix A.4.10.

## 5 EXPERIMENTS

### 5.1 EXPERIMENTAL DETAILS

**Datasets.** We consider three representative datasets: CIFAR-100, Tiny-ImageNet and ImageNet-R with 60k, 100k, 30k training samples and 100, 200, 200 classes, respectively. We follow the setting in Si-Blurry (Moon et al., 2023) in our GCL experiments. Specifically, the *disjoint class ratio* $m$ is set to $50\%$ and the *blurry sample ratio* $n$ is set to $10\%$. We perform 5-session GCL on these three datasets. An extended study of one new class at each session can be found in Appendix A.4.8. Unless specified, the results are obtained on 5 runs with independent seeds. By default, our ISA uses the ImageNet-1k dataset (Deng et al., 2009) of 1000-class large-scale images.

**Baselines.** We consider a variety of CL methods from different categories, including replay-based methods: ER (Rolnick et al., 2019), DER++ (Buzzega et al., 2020), ER-ACE (Caccia et al., 2021), Rainbow Memory (RM) (Bang et al., 2021), and CLIB (Koh et al., 2021); regularization-based methods: LwF (Li & Hoiem, 2018), EWC (Kirkpatrick et al., 2017); and prompt-based methods: L2P (Wang et al., 2022d) and DualPrompt (Wang et al., 2022c). We also compare with the best-performing state-of-the-art prompt-based GCL method to date: MVP (Moon et al., 2023). Continual fine-tuning and linear probing are considered as the lower-bound methods. All the methods share the same frozen pretrained `vit-base-patch16-224` model, except EWC++ and fine-tuning that conducted full parameter tuning on the same pretrained model. Following the model architecture of MVP for a fair comparison, unless specified, we opt for DualPrompt as the baseline method and then add our components. We note that our proposed method is not restricted to specific model designs.

Table 1: Average accuracy (%) with standard deviation of different methods, tested with 5-task CIFAR-100, Tiny-ImageNet, and ImageNet-R.

| Buffer | Method | CIFAR-100 | | Tiny-ImageNet | | ImageNet-R | |
|---|---|---|---|---|---|---|---|
| | | $A_{AUC} \uparrow$ | $A_{Last} \uparrow$ | $A_{AUC} \uparrow$ | $A_{Last} \uparrow$ | $A_{AUC} \uparrow$ | $A_{Last} \uparrow$ |
| 0 | Finetuning | $19.71_{\pm 3.39}$ | $10.42_{\pm 4.92}$ | $15.50_{\pm 0.74}$ | $10.42_{\pm 4.92}$ | $7.51_{\pm 3.94}$ | $2.29_{\pm 0.85}$ |
| | Linear Probe | $49.69_{\pm 6.09}$ | $23.07_{\pm 7.33}$ | $42.15_{\pm 2.79}$ | $21.97_{\pm 6.43}$ | $29.24_{\pm 1.26}$ | $16.87_{\pm 3.14}$ |
| | EWC | $49.51_{\pm 0.52}$ | $52.83_{\pm 2.31}$ | $51.70_{\pm 2.89}$ | $31.04_{\pm 3.12}$ | $31.58_{\pm 1.04}$ | $20.72_{\pm 1.11}$ |
| | LwF | $55.51_{\pm 3.49}$ | $36.53_{\pm 10.96}$ | $49.00_{\pm 1.52}$ | $27.47_{\pm 7.59}$ | $31.61_{\pm 1.53}$ | $20.62_{\pm 3.67}$ |
| | L2P | $57.08_{\pm 4.43}$ | $41.63_{\pm 12.73}$ | $52.09_{\pm 1.92}$ | $35.05_{\pm 5.73}$ | $29.65_{\pm 1.63}$ | $19.55_{\pm 4.78}$ |
| | DualPrompt | $67.07_{\pm 4.16}$ | $56.82_{\pm 3.49}$ | $66.09_{\pm 2.00}$ | $48.72_{\pm 3.41}$ | $40.11_{\pm 1.27}$ | $29.24_{\pm 4.63}$ |
| | MVP | $68.10_{\pm 4.91}$ | $62.59_{\pm 2.38}$ | $68.95_{\pm 1.33}$ | $52.78_{\pm 2.08}$ | $40.60_{\pm 1.21}$ | $31.96_{\pm 3.07}$ |
| | Ours | $\mathbf{80.55}_{\pm 2.17}$ | $\mathbf{80.98}_{\pm 1.08}$ | $\mathbf{80.44}_{\pm 0.80}$ | $\mathbf{74.84}_{\pm 0.64}$ | $\mathbf{50.89}_{\pm 1.03}$ | $\mathbf{43.92}_{\pm 0.37}$ |
| 500 | ER | $65.57_{\pm 4.77}$ | $60.68_{\pm 1.15}$ | $59.46_{\pm 1.81}$ | $40.60_{\pm 2.71}$ | $40.31_{\pm 1.33}$ | $28.85_{\pm 1.43}$ |
| | DER++ | $66.92_{\pm 4.16}$ | $65.63_{\pm 0.72}$ | $61.67_{\pm 1.19}$ | $46.03_{\pm 1.00}$ | $40.32_{\pm 1.08}$ | $31.53_{\pm 1.57}$ |
| | ER-ACE | $69.36_{\pm 3.01}$ | $72.07_{\pm 0.62}$ | $64.52_{\pm 0.78}$ | $56.82_{\pm 0.67}$ | $41.06_{\pm 1.32}$ | $36.59_{\pm 0.52}$ |
| | RM | $40.86_{\pm 3.32}$ | $23.94_{\pm 0.61}$ | $31.96_{\pm 0.80}$ | $7.43_{\pm 0.27}$ | $18.31_{\pm 1.09}$ | $4.14_{\pm 0.18}$ |
| | CLIB | $69.68_{\pm 2.20}$ | $67.16_{\pm 0.72}$ | $60.11_{\pm 1.53}$ | $48.97_{\pm 1.48}$ | $37.18_{\pm 1.52}$ | $29.51_{\pm 0.98}$ |
| | MVP | $76.06_{\pm 4.22}$ | $79.32_{\pm 1.28}$ | $76.52_{\pm 0.73}$ | $65.19_{\pm 0.58}$ | $49.07_{\pm 1.47}$ | $44.17_{\pm 1.72}$ |
| | Ours | $\mathbf{82.37}_{\pm 1.54}$ | $\mathbf{82.27}_{\pm 0.73}$ | $\mathbf{79.08}_{\pm 0.60}$ | $\mathbf{69.91}_{\pm 0.52}$ | $\mathbf{54.72}_{\pm 1.15}$ | $\mathbf{47.48}_{\pm 0.57}$ |
| 2000 | ER | $69.86_{\pm 4.08}$ | $71.81_{\pm 0.69}$ | $66.75_{\pm 1.28}$ | $55.07_{\pm 1.28}$ | $45.74_{\pm 1.35}$ | $38.13_{\pm 0.32}$ |
| | DER++ | $69.42_{\pm 3.65}$ | $65.68_{\pm 0.72}$ | $66.58_{\pm 0.88}$ | $56.81_{\pm 0.65}$ | $42.79_{\pm 1.31}$ | $36.06_{\pm 1.04}$ |
| | ER-ACE | $70.59_{\pm 3.02}$ | $74.75_{\pm 0.19}$ | $66.86_{\pm 0.84}$ | $58.40_{\pm 0.38}$ | $43.62_{\pm 1.31}$ | $40.49_{\pm 0.22}$ |
| | RM | $53.27_{\pm 3.00}$ | $65.51_{\pm 0.55}$ | $47.26_{\pm 1.13}$ | $44.55_{\pm 0.37}$ | $27.88_{\pm 1.29}$ | $24.25_{\pm 0.99}$ |
| | CLIB | $71.53_{\pm 2.61}$ | $72.09_{\pm 0.49}$ | $65.47_{\pm 0.76}$ | $56.87_{\pm 0.54}$ | $42.69_{\pm 1.30}$ | $35.43_{\pm 0.38}$ |
| | MVP | $78.65_{\pm 3.59}$ | $84.42_{\pm 0.44}$ | $80.67_{\pm 0.75}$ | $74.34_{\pm 0.32}$ | $52.47_{\pm 1.45}$ | $50.54_{\pm 2.08}$ |
| | Ours | $\mathbf{83.58}_{\pm 1.72}$ | $\mathbf{85.32}_{\pm 0.25}$ | $\mathbf{82.91}_{\pm 0.47}$ | $\mathbf{76.41}_{\pm 0.33}$ | $\mathbf{57.67}_{\pm 0.72}$ | $\mathbf{53.62}_{\pm 0.68}$ |

Table 2: Ablation studies of the proposed forgetting-aware initial session adaption (ISA-FAM) and non-parametric logit mask (Logit Mask) in MISA.

| Method | Component | | CIFAR-100 | | ImageNet-R | |
|---|---|---|---|---|---|---|
| | ISA-FAM | Logit Mask | $A_{AUC} \uparrow$ | $F_{Last} \downarrow$ | $A_{AUC} \uparrow$ | $F_{Last} \downarrow$ |
| Baseline | | | $67.07_{\pm 4.16}$ | $35.12_{\pm 2.44}$ | $40.11_{\pm 1.27}$ | $43.27_{\pm 6.35}$ |
| Ours | ✓ | | $68.97_{\pm 0.85}$ | $30.01_{\pm 4.14}$ | $41.09_{\pm 1.74}$ | $41.51_{\pm 7.75}$ |
| | | ✓ | $74.84_{\pm 2.99}$ | $11.71_{\pm 1.56}$ | $45.59_{\pm 1.71}$ | $20.84_{\pm 5.49}$ |
| | ✓ | ✓ | $\mathbf{80.55}_{\pm 2.17}$ | $\mathbf{10.35}_{\pm 1.12}$ | $\mathbf{50.89}_{\pm 1.03}$ | $\mathbf{19.91}_{\pm 4.21}$ |

**Implementation Details.** For existing methods, we reuse the implementation of Moon et al. (2023). For MISA, we follow the same training configuration to ensure a fair comparison, e.g., using an Adam optimizer with learning rate 0.005 and batch size 32. In ISA, we use an Adam optimizer with learning rate 0.0001 and batch size 128 to train the prompts in an offline manner for three epochs. For FAM, we split ImageNet-1k into 900 classes for $\mathcal{D}_{id}$ and 100 classes for $\mathcal{D}_{ood}$ without overlap. We randomly sample 10 classes as $\mathcal{D}_{ood}$ to simulate a small-scale downstream task and resample new ones when we iterate over this subset. More aggressive augmentation is applied on $\mathcal{D}_{ood}$ data as we applied double auto-augmentation (Cubuk et al., 2019). For a fair comparison with existing logit masking strategies, we apply our batch-level mask in MISA by default as it does not require a session identifier. More details can be found in Appendix A.1.

**Evaluation Metrics.** We follow the evaluation protocol of Moon et al. (2023), using $A_{AUC}$ and $A_{Last}$ as the main evaluation metrics. $A_{AUC}$ (Koh et al., 2021) measures the performance of anytime inference, with no task labels provided to satisfy the "no test-time oracle" requirement in GCL. $A_{Last}$ is equivalent to the final average accuracy in conventional CL. As we identified forgetting as an essential issue in existing methods, we include $F_{Last}$ as the average forgetting at the end of training. More details on these metrics can be found in Appendix A.2.

## 5.2 BENCHMARK RESULTS

We compare our MISA to existing methods in Tab.1. Our approach consistently outperforms all the other methods by a significant margin, especially when the model is trained *without* rehearsal, i.e., buffer size = 0. This is an extremely challenging case as previous GCL methods often rely on a reply buffer to alleviate forgetting. Specifically, MISA outperforms the previous state-of-the-art MVP by 18.39, 22.06, and 11.96 % points in $A_{Last}$ on CIFAR-100, Tiny-ImageNet, and ImageNet-R datasets, respectively. While $A_{Last}$ focuses on the overall performance at the end of the training, $A_{AUC}$ better exhibits the effectiveness in adaptation to the online data stream. Our approach outperforms MVP

Table 3: Comparison of different masking strategies with different batch sizes. All masking strategies are applied to the same baseline method.

| Mask | Buffer 0 | | | Buffer 500 | | |
|---|---|---|---|---|---|---|
| | 1 | 32 | 64 | 1 | 32 | 64 |
| Seen-Class | $35.41_{\pm0.87}$ | $39.40_{\pm2.72}$ | $37.15_{\pm1.64}$ | $42.68_{\pm1.24}$ | $48.91_{\pm2.00}$ | $45.32_{\pm1.74}$ |
| Learnable | $36.00_{\pm1.54}$ | $39.88_{\pm1.73}$ | $37.74_{\pm1.49}$ | $44.40_{\pm1.97}$ | $44.83_{\pm1.86}$ | $43.49_{\pm2.11}$ |
| Batch-Level | $1.50_{\pm0.04}$ | $45.59_{\pm1.71}$ | $\mathbf{48.41}_{\pm2.14}$ | $34.14_{\pm0.89}$ | $\mathbf{49.00}_{\pm1.46}$ | $47.87_{\pm2.60}$ |
| Session-Level | $\mathbf{44.19}_{\pm0.19}$ | $\mathbf{48.71}_{\pm3.57}$ | $48.19_{\pm2.31}$ | $\mathbf{44.58}_{\pm0.77}$ | $48.99_{\pm2.00}$ | $\mathbf{50.26}_{\pm1.16}$ |

Table 4: $A_{\text{AUC}}$ scores on CIFAR100 and ImageNet-R with prompts obtained by different strategies.

| Method | Logit Mask | Naive ISA | SAM | FAM | CIFAR-100 | ImageNet-R |
|---|---|---|---|---|---|---|
| Baseline | ✓ | | | | $74.84_{\pm2.99}$ | $45.59_{\pm1.71}$ |
| Ours | ✓ | ✓ | | | $77.21_{\pm2.66}$ | $47.71_{\pm1.31}$ |
| | ✓ | ✓ | ✓ | | $79.13_{\pm2.38}$ | $49.99_{\pm1.21}$ |
| | ✓ | ✓ | | ✓ | $\mathbf{80.55}_{\pm2.17}$ | $\mathbf{50.89}_{\pm1.03}$ |

by a margin of 12.45, 11.49, and 10.29 % points in $A_{\text{AUC}}$ on the three datasets, respectively. In particular, MISA beats all conventional CL methods in this measure, which was not achieved by MVP. Furthermore, our approach significantly reduces the variance of the performance with different stochastic classes and data compositions, which confirms its robustness to different realistic learning scenarios.

The integration of the replay buffer can further improve our approach and ensure its standout performance over existing methods in the same configuration. We also note that MVP benefits more from the replay buffer than MISA. We believe it is because MVP suffers from strong forgetting due to their ineffective logit mask. The existence of a replay buffer relieves MVP from this issue. In contrast, MISA better handles the interference of new and old knowledge through our non-parametric logit masking and performs consistently well with or without a replay buffer. Therefore, the replay-independent nature makes MISA a more suitable choice in practice.

## 5.3 FURTHER ANALYSIS

**Ablation Study.** We present the ablation study on CIFAR-100 and ImageNet-R for our proposed components in Tab. 2. The baseline method is overwhelmed by severe forgetting, which prevents the model from accumulating knowledge for good average accuracy, even with forgetting-aware ISA (ISA-FAM). Instead, with the integration of our non-parametric logit masking, forgetting is largely reduced and the performance improves. Finally, without the interference of forgetting, ISA-FAM becomes much more effective. These experiments justify the importance of swift adaption and knowledge retention, as well as their complementary roles in GCL.

**Logit Masking vs Batch Size.** We examine the effectiveness of different logit masking strategies for different batch sizes in Tab. 3. Our session-level masking performs especially well when no replay buffer is available. Unsurprisingly, batch-level mask fails when batch size = 1. This corresponds to an extreme case of online learning where the model receives one sample at a time, which is out of the scope for our paper. In addition, the batch-level mask can quickly recover the performance with a replay buffer. Notably, both our non-parametric masks outperform the learnable mask in most cases.

**Effectiveness of FAM in ISA.** We perform a specific ablation study for our forgetting-aware ISA training strategy, by comparing the impact of prompt parameters with different initialization strategies in Tab. 4. All the methods are equipped with our non-parametric logit mask to overcome the negative impact of strong forgetting. The baseline method utilizes a uniform initialization for prompt parameters. Initializing prompts from a naive ISA, i.e., pretraining with Eq. 2, provides better initialization but lacks considerations for generalizability. Thus, direct integration of SAM can improve performance. Our FAM further boosts the model by enhancing the robustness to distribution shift to maintain the generalizability in downstream GCL, as highlighted by the $A_{\text{AUC}}$ score. Note that both SAM and FAM are equipped with prompt augmentation to enable flatness-aware minimization, whose ablation study can be found in Appendix A.4.4.

Table 5: Validation of transferability of our proposed components on existing prompt-based methods on ImageNet-R. For ISA, both methods reuse the same prompts as our approach.

| Baseline | ISA-FAM & Logit Mask | $A_{\text{AUC}}$ ↑ | $A_{\text{Last}}$ ↑ |
|---|---|---|---|
| L2P |  | $29.42_{\pm1.46}$ | $20.46_{\pm4.17}$ |
|  | ✓ | $\mathbf{34.11}_{\pm0.36}$ | $\mathbf{27.86}_{\pm1.11}$ |
| MVP |  | $40.60_{\pm1.21}$ | $31.96_{\pm3.07}$ |
|  | ✓ | $\mathbf{43.01}_{\pm1.26}$ | $\mathbf{33.52}_{\pm3.38}$ |

Table 6: Performance of pre-trained checkpoint *not* overlapping with CIFAR-100 or Tiny-ImageNet.

| Method | CIFAR-100 | | Tiny-ImageNet | | ImageNet-R | | NCH | |
|---|---|---|---|---|---|---|---|---|
|  | $A_{\text{AUC}}$ | $A_{\text{Last}}$ | $A_{\text{AUC}}$ | $A_{\text{Last}}$ | $A_{\text{AUC}}$ | $A_{\text{Last}}$ | $A_{\text{AUC}}$ | $A_{\text{Last}}$ |
| EWC | $41.5_{\pm6.6}$ | $29.4_{\pm3.3}$ | $40.9_{\pm2.3}$ | $23.3_{\pm5.5}$ | $31.8_{\pm1.3}$ | $19.9_{\pm6.2}$ | $50.0_{\pm12.4}$ | $32.6_{\pm9.7}$ |
| DualPrompt | $47.2_{\pm2.8}$ | $42.7_{\pm4.6}$ | $48.2_{\pm1.1}$ | $31.2_{\pm2.7}$ | $33.7_{\pm1.0}$ | $26.7_{\pm0.8}$ | $60.6_{\pm11.7}$ | $49.4_{\pm7.8}$ |
| MVP | $48.8_{\pm5.1}$ | $35.5_{\pm5.1}$ | $46.5_{\pm2.0}$ | $26.8_{\pm4.5}$ | $35.4_{\pm1.4}$ | $24.2_{\pm4.4}$ | $60.4_{\pm10.1}$ | $43.5_{\pm11.6}$ |
| Ours | $\mathbf{55.7}_{\pm1.9}$ | $\mathbf{53.8}_{\pm3.6}$ | $\mathbf{56.5}_{\pm2.0}$ | $\mathbf{47.4}_{\pm1.7}$ | $\mathbf{41.2}_{\pm1.0}$ | $\mathbf{35.9}_{\pm0.9}$ | $\mathbf{71.2}_{\pm3.3}$ | $\mathbf{61.8}_{\pm7.4}$ |

**Transferability of MISA.** Our approach is designed to be general and plug-in for prompt-based CL methods. To demonstrate this transferability, we evaluate our approach with two other methods: L2P and MVP. The former is a representative prompt-based method for conventional CL, and the latter is the previous state-of-the-art prompt-based method for GCL. As shown in Tab. 5, both methods can benefit from our design. Note that these methods share the same initialized prompts parameters as our approach, without performing additional ISA. The full results can be found in Appendix A.4.1.

**Knowledge Overlap Analysis.** This analysis aims to reduce the potential knowledge overlap between pretraining data and downstream GCL. Specifically, we consider a particular pretrained checkpoint (Kim et al., 2022), which removes 389 classes from ImageNet-1k that are similar to CIFAR-100 or Tiny-ImageNet. We repeat our experiments on this subset of ImageNet-1k to exclude information leaks to either the backbone or prompt parameters from ISA. Moreover, we included a completely out-of-distribution dataset NCH (Kather et al., 2018) from the medical image domain to showcase the transferability of our learned prompts. As shown in Tab. 6, our approach still achieves substantial improvements over all baselines (see details in Appendix A.4.2).

**Memory and Computation Analysis.** We analyze in Tab.7 the complexity of representative methods. Specifically, we report the average execution time for each method to learn from one batch on 1 RTX A5000 GPU. Our approach brings a significant performance gain without introducing additional parameters or increased execution time. Combined with the replay-independent nature, our MISA has further satisfied the requirements of GCL in terms of constant memory.

Table 7: Comparison of resource overheads with the same machine and configuration.

| Method | # Param. | Time | Accuracy |
|---|---|---|---|
| DualPrompt | 637k | 1.31s | 67.07 |
| MVP | 639k | 2.75s | 68.10 |
| Ours | 637k | 1.32s | **80.55** |

**CL-Relevant Hyperparameters.** We argue that the hyperparameters of MVP selected with validation data from a static view is contradictory to the stochastic nature of data distribution. In contrast, our approach does not require CL-relevant hyperparameters, which is an important advantage for facilitating the use of our components in other methods or scenarios such as GCL.

## 6 CONCLUSION

In this paper, we advance prompt-based methods in the challenging GCL setting with our novel approach MISA, which consists of a forgetting-aware initial session adaption to improve learning efficiency and generalizability of the prompt parameters, and a non-parametric logit mask to alleviate forgetting of the output layers. Our extensive experiments show the effectiveness of each component and the significant performance gain compared to existing methods, setting a new state of the art for GCL. Moreover, our approach is compatible with other prompt-based CL methods, which can seamlessly benefit from our components. Finally, our approach overcomes the need for CL-relevant hyperparameters and is independent of replay data, enabling robustness and flexibility for CL applications.

REPRODUCIBILITY STATEMENT

To ensure the reproducibility of our work, we have provided comprehensive details about the experimental setup in Section 5, including datasets used, baseline methods, implementation details and evaluation metrics. More details about the implementation can be found in Appendix A.1. All code, models, and configuration files required to replicate our results are made publicly available in a repository.

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

## A  APPENDIX

### A.1  IMPLEMENTATION DETAILS

We follow the implementation of Moon et al. (2023) for a fair comparison with existing methods. Specifically, all methods use a pretrained ViT checkpoint *ViT-B/16* (Dosovitskiy et al., 2020). We freeze the backbone, learning only the prompt parameters and the classification head. As such, the total number of parameters is around 86M, whereas the trainable one is around 0.6M. We equip our approach with the same memory buffer as DualPrompt in the implementation of Moon et al. (2023) with size 500 and 2000, if applicable.

For forgetting-aware optimization, we use the first 900 classes of ImageNet-1k as $\mathcal{D}_{id}$ for initializing the parameters and the last 100 classes for $\mathcal{D}_{ood}$ to calculate perturbation. During training, we randomly sample 10 classes as actual $\mathcal{D}_{ood}$. Since this subset is smaller than $\mathcal{D}_{id}$, whenever we go through the entire small subset, we will resample the other 10 classes. We will repeat this process until the end of training. To apply a more aggressive data augmentation on $\mathcal{D}_{ood}$ data, we applied two times `AutoAugment` to augment the image, while for $\mathcal{D}_{id}$ data, we only applied it once.

For prompt augmentation, we use a two-layer MLP with projection dimensions $(768, 96)$ and $(96, 768)$, which corresponds to a down-sampling rate of $1/8$ for the feature dimension $D$. We empirically found it more effective than up-sampling. We add LayerNorm (Ba et al., 2016) and ReLU (Agarap, 2018) activation in between these two layers. Once the ISA is finished, this MLP is discarded.

### A.2  EVALUATION METRICS

We start from conventional evaluation metrics in CL. Let $R_{i,j}$ be the evaluation score of the model after session $i$ with respect to the data in the session $j$. We then have an evaluation matrix $R$ such that the column $R_j$ shows the history of evaluation after each session with respect to the data in the session $j$. The final accuracy $A_{\text{Last}}$ can be defined as:

$$A_{\text{Last}} = \frac{1}{T} \sum_{i=1}^{T} R_{T,i}, \tag{9}$$

and the forgetting can be defined as:

$$F_{\text{Last}} = \frac{1}{T} \sum_{i=1}^{T} (\max(R_j) - R_{T,i}), \tag{10}$$

where $A_{\text{Last}}$ is the higher the better and $F_{\text{Last}}$ is the lower the better.

$A_{\text{AUC}}$ is the anytime inference metric proposed by Koh et al. (2021) to evaluate the GCL performance. Here, the evaluation is not performed at the end of each session but whenever the model observes a number of $n$ samples. Let the total number of evaluations performed be $L$ and $l$ be the time-step for evaluation. We have:

$$A_{\text{AUC}} = \frac{1}{L} \sum_{l=1}^{L} R_{l,l}, \tag{11}$$

where $R_{l,l}$ shows the evaluation score of the model at the time-step $l$ with respect to the data it has observed until time-step $l$.

### A.3  SI-BLURRY AS A REALIZATION OF GENERALIZED CLASS INCREMENTAL LEARNING

Following the definition of generalized class incremental learning (GCIL) in Mi et al. (2020), we reformulate the Si-Blurry scenario in this GCIL framework. Let $\mathbb{C}$ be the set of available classes whose cardinality is $N = |\mathbb{C}|$. Let $\{\mathbb{C}^D, \mathbb{C}^B\}$ be a partition of $\mathbb{C}$ where $\mathbb{C}^D$ represents disjoint classes and $\mathbb{C}^B$ represents blurred classes. $m$ is the *disjoint class ratio* such that $m = |\mathbb{C}^D|/|\mathbb{C}|$. Let $\mathbb{X}$ be the set of available samples, we thus have: $\mathbb{X}^D = \{x_i : y_i \in \mathbb{C}^D\}$ and $\mathbb{X}^B = \{x_i : y_i \in \mathbb{C}^B\}$, where $y_i$ is the class label of the sample $x_i$. Let $T$ be the total number of learning sessions. $\{\mathbb{C}_t^D\}_{t=1...T}$ and $\{\mathbb{C}_t^B\}_{t=1...T}$ are then a partition of $\mathbb{C}^D$ and $\mathbb{C}^B$ to allocate classes for each session,

Table 8: Validation of transferability of our proposed components on existing prompt-based methods. We perform experiments on 5-task ImageNet-R, averaged over 5 runs.

| Baseline | Component | | ImageNet-R | |
|---|---|---|---|---|
| | Init-Adapt | Logit-Mask | $A_{\text{AUC}} \uparrow$ | $A_{\text{Last}} \uparrow$ |
| L2P | | | $29.42_{\pm 1.46}$ | $20.46_{\pm 4.17}$ |
| | ✓ | | $31.72_{\pm 1.36}$ | $21.09_{\pm 3.87}$ |
| | | ✓ | $31.84_{\pm 2.99}$ | $25.88_{\pm 0.83}$ |
| | ✓ | ✓ | $\mathbf{34.11}_{\pm 0.36}$ | $\mathbf{27.86}_{\pm 1.11}$ |
| MVP | | | $40.60_{\pm 1.21}$ | $31.96_{\pm 3.07}$ |
| | ✓ | | $41.12_{\pm 1.77}$ | $32.05_{\pm 1.32}$ |
| | | ✓ | $40.98_{\pm 0.85}$ | $32.31_{\pm 3.32}$ |
| | ✓ | ✓ | $\mathbf{43.01}_{\pm 1.26}$ | $\mathbf{33.52}_{\pm 3.38}$ |

respectively. In Si-Blurry, the number of classes for each session is not uniform across tasks as the number is randomly sampled. For disjoint data, we have $\mathbb{X}_t^D = \{x_i : y_i \in \mathbb{C}_t^D\}$. Let $\{\tilde{\mathbb{X}}^B, \bar{\mathbb{X}}^B\}$ be a partition of $\mathbb{X}^B$, with a *blurry sample ratio* $n = |\tilde{\mathbb{X}}^B|/|\mathbb{X}^B|$. $\bar{\mathbb{X}}^B$ is a normal set and $\tilde{\mathbb{X}}^B$ is the blurred set. $\bar{\mathbb{X}}^B$ will be divided into sessions as $\bar{\mathbb{X}}_t^B = \{x_i : x_i \in \bar{\mathbb{X}}^B, y_i \in \mathbb{C}_t^B\}$. In a blurred boundary setting, $\tilde{\mathbb{X}}^B$ is non-empty and will be shuffled and randomly distributed to different sessions as $\{\tilde{\mathbb{X}}_t^B\}$. The final set of data available to the session $t$ is then $\mathbb{X}_t = \mathbb{X}_t^D \cup \bar{\mathbb{X}}_t^B \cup \tilde{\mathbb{X}}_t^B$. Accordingly, the classes available for the session is $\mathbb{C}_t = \mathbb{C}_t^D \cup \mathbb{C}_t^B \cup \tilde{\mathbb{C}}_t^B$ where $\forall x_i \in \tilde{\mathbb{X}}_t^B, y_i \in \tilde{\mathbb{C}}_t^B$. With this formulation, we revisit the properties of GCIL defined in Mi et al. (2020) to show that Si-Blurry can be seen as a realization of GCIL in an online learning paradigm.

**Property 1:**   The number of classes could differ in each session. We have:

$$\forall i, j, i \neq j, P(|\mathbb{C}_i| \neq |\mathbb{C}_j|) > 0, \tag{12}$$

where $P(\cdot)$ is the probability.

**Property 2:**   Classes could appear in different sessions. We have:

$$\forall i, j, i \neq j, P(\mathbb{C}_i \cap \mathbb{C}_j \neq \emptyset) > 0, \tag{13}$$

**Property 3:**   The number of samples for each class could be different in one session. For a session $t$, we have:

$$\forall i, j, i \neq j, P(|\mathbb{X}_t^i| \neq |\mathbb{X}_t^j|) > 0, \tag{14}$$

where $|\mathbb{X}_t^i|$ is the number of samples for class $i$ in session $t$.

As GCIL in Mi et al. (2020) has no explicit constraint on the number of iterations allowed for learning, Si-Blurry features the more complex and realistic setting with the one-pass requirement. In summary, Si-Blurry can be seen as a realization of GCIL of Mi et al. (2020) with an online learning paradigm.

## A.4   Additional Experiments

### A.4.1   Transferability

We include the full table for the validation of the transferability of our proposed components on existing prompt-based methods for the effectiveness of each component, as shown in Tab. 8.

### A.4.2   Knowledge overlap analysis

To ensure that there is no overlapping of data with ImageNet, Kim et al. (2022) manually removed 389 classes from the original 1000 classes in ImageNet that are similar/identical to the classes in CIFAR-10, CIFAR-100, or Tiny-ImageNet. They pre-trained the network with the remaining subset of 611 classes of ImageNet and released the checkpoint for public research purposes.

While the list of the 389 classes is not publicly available, we reached out to the authors for the list and performed additional ISA to obtain prompt parameters that are initialized without knowledge

Table 9: Validation of ISA for different types of prompts. All methods are equipped with our logit mask to exclude the impact of strong forgetting.

| Baseline | | Prompt Type | | ImageNet-R | |
|---|---|---|---|---|---|
| | | G-prompt | E-prompt | $A_{\text{AUC}}$ ↑ | $A_{\text{Last}}$ ↑ |
| DualPrompt | | ✓ | | $47.84_{\pm 1.49}$ | $40.82_{\pm 0.73}$ |
| | | | ✓ | $48.40_{\pm 1.20}$ | $42.66_{\pm 0.34}$ |
| | | ✓ | ✓ | $\mathbf{50.89}_{\pm 1.03}$ | $\mathbf{43.92}_{\pm 0.37}$ |

Table 10: CL experiments on ImageNet-R with prompts obtained by different strategies. We use our logit masking for all to exclude the impact of strong forgetting.

| Method | | Logit-Mask | Naive ISA | SAM | Augmentation | $A_{\text{AUC}}$ |
|---|---|---|---|---|---|---|
| Baseline | | ✓ | | | | $45.59_{\pm 1.71}$ |
| Ours | | ✓ | ✓ | | | $47.64_{\pm 1.24}$ |
| | | ✓ | ✓ | ✓ | | $47.71_{\pm 1.31}$ |
| | | ✓ | ✓ | ✓ | ✓ | $\mathbf{49.99}_{\pm 1.21}$ |

overlap. These additional experiments ensure that the improvements of our proposed components are not from a leak of information from the pretraining data.

### A.4.3 SAM WITH PROMPT-TUNING

Although SAM has been shown to be effective in improving the generalizability of CL models, it works most of the time with full-parameter tuning. The effectiveness with parameter-efficient tuning methods, i.e., a frozen pretrained backbone and a few learnable parameters, remains unclear. We first conducted a toy example by applying SAM with a pretrained ViT backbone and prompt parameters in a two-task conventional CL setting. The evaluation on the first task shows the convergence of the model, whereas the evaluation on the second task is a direct verification of the model's generalizability.

As shown in Fig. 3, directly integrating the SAM optimizer into prompt-based tuning cannot work well, or even hinders the model's convergence. We speculate it is because the simplicity and scarcity of the prompts make them incapable of modifying the loss surface of the pretrained model. Moreover, the imposed flatness constraint prevents the loss from converging to the original sharp minima. Based on this analysis, we further conduct experiments by de-freezing more parameters to the training. Specifically, $0.3M$ refers to tuning the prompts $\boldsymbol{p}_e$ and the output layer $f_c$, while 7M and 14M introduce additional parameters from the last one or two self-attention blocks of $f_r$, respectively. We observe that when more parameters are available for training, the SAM begins to show its power. This experiment confirms that SAM has difficulty when applied to a large frozen pretrained model and a few learnable prompt parameters.

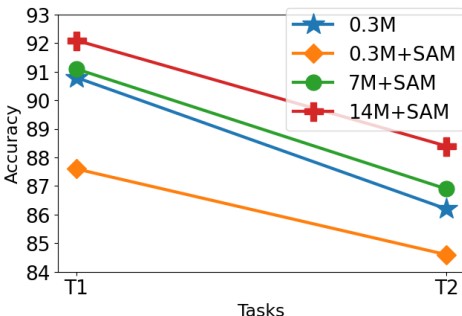

Figure 3: Toy example of implementing SAM with prompt tuning. We performed 2-task CL with randomly sampled 50 classes from ImageNet-1K. The first task is optimized with SAM and the second task uses a standard Adam optimizer. 0.3M, 7M, and 14M represent the number of learnable parameters, and additional parameters are from the last layers of the pretrained backbone $f_r$. SAM works better with more parameters becoming learnable.

Table 11: Validation of the effectiveness of ISA with respect to the training iterations.

| Baseline | | Epoch | | ImageNet-R | |
| --- | --- | --- | --- | --- | --- |
| | | | | $A_{\text{AUC}} \uparrow$ | $A_{\text{Last}} \uparrow$ |
| DualPrompt | | 1 | | $48.14_{\pm 1.26}$ | $42.40_{\pm 0.57}$ |
| | | 2 | | $49.79_{\pm 1.11}$ | $43.33_{\pm 0.52}$ |
| | | 3 | | $\mathbf{49.99}_{\pm 1.21}$ | $\mathbf{43.48}_{\pm 0.43}$ |
| | | 4 | | $49.87_{\pm 1.31}$ | $43.22_{\pm 0.85}$ |

Table 12: Validation of the effectiveness of downstream task adaptation.

| Baseline | | Adaptation | | ImageNet-R | |
| --- | --- | --- | --- | --- | --- |
| | | | | $A_{\text{AUC}} \uparrow$ | $A_{\text{Last}} \uparrow$ |
| DualPrompt | | | | $49.99_{\pm 1.21}$ | $43.48_{\pm 0.43}$ |
| | | $\checkmark$ | | $\mathbf{50.16}_{\pm 1.22}$ | $\mathbf{43.56}_{\pm 0.42}$ |

### A.4.4 ABLATION STUDY OF PROMPT AUGMENTATION

We show in Tab. 10 that direct integration of SAM shows only marginal performance gain, as SAM struggles to work with prompt-tuning. In the end, with our prompt augmentation technique, we observe an additional boost to the performance.

### A.4.5 IMPACT OF PROMPTS (G-PROMPT AND E-PROMPT)

As our baseline method (Wang et al., 2022c) is composed of two types of prompts: G-prompt, which is for capturing general information, and E-prompt, which is for capturing task-specific information. We conduct an ablation study in Tab. 9 on these two types of prompts to validate the effectiveness of each type of prompt.

### A.4.6 IMPACT OF TRAINING EPOCHS

Our ISA involves a joint offline training on the large-scale dataset. The training only lasts for 3 epochs. We show in Tab. 11 that the prompt parameters the best generalizable warm-up at the end of epoch 3. Further training brings no significant improvements.

### A.4.7 IMPACT OF DOWNSTREAM CL ADAPTATION

Our approach directly uses the prompts obtained from ISA to downstream tasks. However, in the cases where there is a significant domain gap between the ISA dataset and the downstream CL dataset, it is preferable to apply a dedicated domain adaptation module to compensate for this mismatch. We found that a simple shift-and-scale method (Lian et al., 2022) can help the model better adapt the downstream tasks without hindering the knowledge in the ISA prompts, as shown in Tab. 12. We leave this adaption as an interesting future direction since it was not the main focus of this work.

### A.4.8 CASE STUDY OF ONE CLASS AT A TIME GCL

We conducted a case study of an extreme case where each new task comprises only one class. Note that the tasks are not strictly disjoint, as the data stream can still contain data from previous tasks due to GCL's blurry task boundary requirements. Typically, any class-incremental scenario can be decomposed to one-class increments, without assuming the number of classes in each new task. This challenging setting demands the model to swiftly adapt and learn new knowledge about a new single class without forgetting previous knowledge that is underrepresented in the current data stream. The results are presented in Tab. 13. Specifically, we conducted 100-task CIFAR-100 and 200-task ImageNet-R experiments with 5 random seeds. We note that our method still outperforms existing methods by a substantial margin.

Table 13: Performance of different methods in GCL with one incremental class at a time.

| Method | ImageNet-R | | CIFAR-100 | |
|---|---|---|---|---|
| | $A_{\text{AUC}}$ | $A_{\text{Last}}$ | $A_{\text{AUC}}$ | $A_{\text{Last}}$ |
| EWC | $2.3_{\pm 1.1}$ | $3.1_{\pm 1.6}$ | $68.1_{\pm 0.4}$ | $64.5_{\pm 1.4}$ |
| DualPrompt | $25.6_{\pm 5.6}$ | $31.2_{\pm 1.6}$ | $80.4_{\pm 0.4}$ | $79.1_{\pm 0.2}$ |
| MVP | $4.1_{\pm 1.4}$ | $5.2_{\pm 4.4}$ | $78.7_{\pm 0.1}$ | $76.2_{\pm 0.2}$ |
| Ours | $\mathbf{31.1}_{\pm 0.6}$ | $\mathbf{42.7}_{\pm 0.7}$ | $\mathbf{85.3}_{\pm 0.1}$ | $\mathbf{83.7}_{\pm 0.3}$ |

### A.4.9 EXPERIMENTS OF PARTIALLY OR COMPLETELY UNAVAILABLE PRETRAINING DATASET.

Although our ISA experiment was based on the same pretrained dataset as the backbone, this is not a strict requirement. The corresponding pretraining data might be partially or completely unavailable regarding large pretrained vision, vision-language, and language models. In these cases, the ISA should be able to learn robust and transferable general knowledge for the prompt parameters to deal with the potential knowledge mismatch with the backbone. We conduct experiments with two cases: (1) the pretrained data is partially available, (2) the pretrained data is completely unavailable. For the former, we use a subset (ImageNet-100, denoted at IN-100) of the pretraining dataset (ImageNet-1k, denoted as IN-1k), and for the latter, we use ImageNet-1k as the ISA dataset whereas the backbone (CLIP-ViT) was trained on a completely unrelated dataset, i.e., YFCC100M (Thomee et al., 2016). Additionally, we highlight that general and diverse ISA datasets are beneficial for the prompt parameters to capture transferable knowledge. To showcase this, we also conducted ISA on a fine-grained dataset, i.e., CUB200 (Wah et al., 2011), where we observed a clear performance drop. The evaluation was performed with 5-task GCL in ImageNet-R with 5 random seeds. The results are in Tab. 14.

### A.4.10 ANALYSIS OF OUR LOGIT MASKING STRATEGY.

In this section, we compare our non-parametric logic masking with that of ER-ACE Caccia et al. (2021). Although they seem similar, they differ in their use cases and objectives. We follow the notations of ER-ACE in this analysis.

First of all, the mask of ER-ACE is replay-dependent, whereas ours is replay-independent. And we are primarily interested in the case where we do not use a replay buffer.

The second difference lies in the objective of the mask. Although both masks deal with the co-existence of current-class data $\mathbb{X}^{cur}$ and old-class data in one mini-batch, the nature of the data is different. In ER-ACE, old-class data $\mathbb{X}^{bf}$ is sampled from a replay buffer, whereas in our case, old-class data $\mathbb{X}^{old}$ is the new data of old classes from the blurry task boundaries. Let $\mathbb{C}$ be the class in data batch $\mathbb{X}$, and $\mathbb{C}_{seen}$ denote all the classes the model has seen. Thus $\mathbb{C}_{bf} \cup \mathbb{C}_{cur} \subseteq \mathbb{C}_{seen}$. For ER-ACE, they applied $\mathbf{1}_{\mathbb{C}_{cur}}$ for $\mathbb{X}^{cur}$ and $\mathbf{1}_{\mathbb{C}_{seen}}$ for $\mathbb{X}^{bf}$, with $\mathbf{1}_{\mathbb{C}}$ a binary vector masking out classes not in $\mathbb{C}$. The idea was to prevent the gradient update of $\mathbb{X}^{cur}$ from interfering with representations of previously seen classes. This is a rather conservative strategy as the idea is to prevent old-class representation from being changed. Instead, since our $\mathbb{X}^{old}$ are new data of old classes, unlike $\mathbb{X}^{bf}$, we can opt for a more proactive strategy to encourage representation update of $\mathbb{C}_{cur}$ and $\mathbb{C}_{old}$ at the same time. Thus our mask to $\mathbb{X}^{cur} \cup \mathbb{X}^{old}$ is $\mathbf{1}_{\mathbb{C}_{cur} \cup \mathbb{C}_{old}}$. Similarly, we exclude other classes that have not been seen in the current mini-batch to avoid interference with their representations. The overall optimization objective of ER-ACE is

$$L_{ce}(X^{bf}, Y^{bf}|C_{seen}) + L_{CE}(X^{cur}, Y^{cur}|C_{cur}),$$

whereas ours is

$$L_{ce}(X^{old} \cup X^{cur}, Y^{old} \cup Y^{cur}|C_{old} \cup C_{cur}),$$

where $L_{ce}(\mathbb{X}, \mathbb{Y}|\mathbb{C})$ is the cross-entropy loss with $\mathbf{1}_{\mathbb{C}}$ as the binary mask on the data $\mathbb{X}, \mathbb{Y}$.

Table 14: Experiments of partially or completely unavailable pretraining data for ISA.

| Method | PTM(IN-1k)+ISA(IN-100) | | PTM(YFCC100M)+ISA(IN-1k) | | PTM(IN-1k)+ISA(CUB200) | |
|---|---|---|---|---|---|---|
| | $A_{\text{AUC}}$ | $A_{\text{Last}}$ | $A_{\text{AUC}}$ | $A_{\text{Last}}$ | $A_{\text{AUC}}$ | $A_{\text{Last}}$ |
| EWC | $31.5_{\pm 1.0}$ | $20.7_{\pm 1.1}$ | $30.6_{\pm 1.3}$ | $22.2_{\pm 3.8}$ | $31.5_{\pm 1.0}$ | $20.7_{\pm 1.1}$ |
| DualPrompt | $40.1_{\pm 1.2}$ | $29.2_{\pm 4.6}$ | $37.9_{\pm 0.9}$ | $30.3_{\pm 0.2}$ | $40.1_{\pm 1.2}$ | $29.2_{\pm 4.6}$ |
| MVP | $40.6_{\pm 1.2}$ | $31.9_{\pm 3.0}$ | $34.0_{\pm 1.3}$ | $25.0_{\pm 4.4}$ | $40.6_{\pm 1.2}$ | $31.9_{\pm 3.0}$ |
| Ours | $\mathbf{49.5}_{\pm 1.3}$ | $\mathbf{43.0}_{\pm 0.6}$ | $\mathbf{49.3}_{\pm 1.2}$ | $\mathbf{42.3}_{\pm 0.9}$ | $\mathbf{43.4}_{\pm 1.9}$ | $\mathbf{36.6}_{\pm 1.6}$ |

