# OpenReview forum: "Advancing Prompt-Based Methods for Replay-Independent General Continual Learning"
_ICLR.cc/2025/Conference — ICLR 2025 Poster_

### Official Review · Reviewer_1W4v · 2024-10-29

**Soundness:** 3
**Presentation:** 4
**Contribution:** 3
**Rating:** 8
**Confidence:** 5

**Summary:**

The paper proposes a prompt-based method for General Continual Learning (GCL), an incremental setting where task boundaries between consecutive tasks are not rigid, allowing certain classes to reappear in later tasks. GCL represents a challenging yet realistic scenario, as the absence of clear boundaries limits the applicability of many algorithms and techniques, particularly those relying on model expansion. The authors leverage two key strategies: (1) refining the pre-trained model’s initialization to improve resilience to distribution shifts, thereby preserving performance; and (2) using masking techniques during cross-entropy loss computation, selectively based on the classes present in each batch. Experimental results indicate strong, promising performance relative to existing solutions, complemented by extensive ablation studies.

**Strengths:**

- The writing is brilliant. It is a pleasure to review papers like this one.
- The idea behind FAM is noteworthy and represents the main contribution of this work.
- The experiments are extensive and cover all that is needed to understand the proposed approach.

**Weaknesses:**

I have only a few minor questions and suggestions.

1. While reading Sec. 4.1, I was initially puzzled by the optimization problem outlined in Eq. 2. Specifically, I questioned the advantage of fine-tuning prompts using the same data employed during pre-training. Since the gradients should be near zero around the pre-training weights, this raised some concerns about the utility of prompts. The rationale becomes clearer a few lines later; however, to improve readability, I suggest that the authors provide some context earlier in Sec. 4.1.

2. While reading Sec. 4.3, I began to wonder if FAS could also be effectively applied to standard class-incremental learning. The idea does not seem strictly constrained to the GCL setting.

3. In reviewing Eqs. 5 and 6, I noticed similarities with Meta-Learning, particularly Meta Agnostic Meta Learning (MAML). These equations suggest an optimization setup where some data is used to train the model (inner loop) and other data is used for “differentiable” evaluation (outer loop). I believe that discussing this connection in the main paper would add value, but I’d like to hear the authors' perspective on this.

4. The prompt augmentation is the only aspect that doesn’t fully convince me. Even with an MLP layer, the resulting complexity seems comparable to that of a straightforward learnable prompt. Perhaps the MLP affects the training trajectory, introducing a smoothing effect across iterations. In my experience, concatenation-based prompting strategies tend to be less effective, though this may not be due to reduced parameters or complexity. Instead, I suspect it depends on how these prompts are incorporated into the pre-trained model. I recommend that the authors test their approach with [a] as a fine-tuning strategy (i.e., using addition instead of concatenation).

[a] Liu, H., Tam, D., Muqeeth, M., Mohta, J., Huang, T., Bansal, M., & Raffel, C. A. (2022). Few-shot parameter-efficient fine-tuning is better and cheaper than in-context learning. Advances in Neural Information Processing Systems, 35, 1950-1965.

5. The masking strategy used in this work is identical to that of ER-ACE (ICLR 2022). I think the paper should be more transparent about this similarity.

**Questions:**

See section above.

---

> ### Author Response · Authors · 2024-11-19
> **Authors' Response (Part I)**
>
> Thank you for the valuable comments. Below, we provide a point-to-point response to these comments and summarize the corresponding revisions. We would be happy to address any further questions.
>
> ### Q1: I suggest that the authors provide some context earlier in Sec. 4.1.
>
> As suggested, we have added more explanations earlier in Sec. 4.1 to improve its  (see line 225). Recall that the prompt parameters and the classifiers are both randomly initialized, so the gradient for these tunable parameters in Eq. 2 will not be near zero. In addition, the backbone was pretrained **without** prompt parameters, which means that the additional prompt parameters need to be learned to actively capture the information flow in the frozen backbone. This was entangled with continually learning new tasks in previous methods, whereas our ISA disentangles this in the warm-up stage (i.e., our ISA) before any downstream CL tasks.
>
> ### Q2: Effectiveness of forgetting-aware initial session adaption (ISA-FAM) in standard class-incremental learning.
>
> We have evaluated this in our preliminary experiments, where the benefits of ISA-FAM for prompt-based methods are specific to GCL (i.e., online data streams and blurry task boundaries) rather than to standard class-incremental learning (i.e., offline training of each task with explicit boundaries). For example, DualPrompt on CIFAR-100 is improved by 13.48\% with the former, whereas by less than 1\% with the latter. This is because standard class-incremental learning allows for adequate training of prompt parameters and adequate differentiation of task-specific knowledge, enabling prompt-based methods to overcome the poor learning capacity of visual prompts and achieve state-of-the-art performance. However, the specific challenges of GCL significantly compromise the advantages of existing prompt-based methods, leading to poor initial performance and limited generalizability, which motivate our design of ISA-FAM. We have added this discussion in our revised manuscript (see lines 237-241).
>
> ### Q3: Connections of Eqs.5/6 and Meta Agnostic Meta Learning (MAML).
>
> Thank you for pointing out this interesting connection. We agree that Eqs. 5/6 have a form similar to MAML, suggesting that the proposed forgetting-aware minimization promotes the generalizability of forgetting-sensitive directions in a data-driven manner (i.e., by constructing pseudo CL tasks). We have added this in our revised manuscript (see lines 302-304).
>
> ### Q4: Using addition instead of concatenation of prompt parameters.
>
> Thank you for pointing out this alternative implementation $(IA)^3$ as the parameter-efficient fine-tuning method. We have empirically investigated its effectiveness on the DualPrompt backbone. In the following table, we present the evaluation accuracy of different methods at the end of each session in a 5-session GCL on ImageNet-R.
>
> We first consider a shared $(IA)^3$ across all sessions. It is noteworthy that the performance in early sessions (e.g., Session 1) is significantly higher than that of prompt concatenation, which confirms the effectiveness of this method. Since $(IA)^3$ is not a CL method, there is no specific design for parameter expansion or forgetting. It is expected to see performance drop due to forgetting. We further incorporate $(IA)^3$ with a parameter expansion, i.e., introduce a new set of parameters for each session,  and a key-query mechanism to select the corresponding set of parameters during inference time. However, we observe that with this simple adaptation to GCL (parameter expansion and key-match strategy), the performance gain compared to prompt concatenation diminished (last row of the table). We believe that there must be a more effective and efficient way to adapt $(IA)^3$ to a CL setting. This is an interesting and promising future direction in the research of prompt-based methods in GCL.
>
>
> | Method     | Prompt technique             | Sesssion 1 | session 2 | Session 3 | Session 4 | Session 5 |
> |:---------- |:---------------------------- |:---------- |:--------- |:--------- |:--------- |:--------- |
> | DualPrompt | Concatenation                | 60.0       | 40.9      | 36.7      | 24.8      | 28.8      |
> | DualPrompt | $(IA)^3$ (shared)    | 72.3       | 50.9      | 42.3      | 23.8      | 31.2      |
> | DualPrompt | $(IA)^3$ + expansion + key match | 51.4       | 32.4      | 27.8      | 22.6      | 25.6      |

---

> > ### Author Response · Authors · 2024-11-19
> > **Authors' Response (Part II)**
> >
> > ### Q5: Comparison of masking strategy with ER-ACE (ICLR 2022).
> >
> > We would respectfully point out that our non-parametric logic masking is different from the masking strategy of ER-ACE.
> >
> > First of all, the mask of ER-ACE is replay-dependent whereas ours is replay-independent. And we are primarily interested in the case where we do not use a replay buffer.
> >
> > The second difference lies in the objective of the mask. Although both masks deal with the co-existence of current-class data $X_{cur}$ and old-class data in one mini-batch, the nature of the data is different. In ER-ACE, old-class data $X_{bf}$ is sampled from a replay buffer whereas in our case, old-class data $X_{old}$ is the new data of old classes from the blurry task boundaries.
> >
> > Let $C$ be the class that appeared in data batch X, and $C_{seen}$ denote all the classes the model has seen. Thus we have $C_{cur} \cup C_{bf} \subseteq C_{seen}$ and $C_{cur} \cup C_{old} \subseteq C_{seen}$. For ER-ACE, they applied $1_{C_{cur}}$  for $X_{cur}$ and $1_{C_{seen}}$ for $X_{bf}$, with $1_{C}$ a binary vector masking out classes not in $C$. The idea was to prevent gradient update of $X_{cur}$ interfering representations of previously seen classes. This is a rather conservative strategy as the idea is to prevent old-class representation from being changed. Instead, since our $X_{old}$ are new data of old classes, unlike $X_{bf}$, which might contain useful knowledge of the old classes, we can opt for a more proactive strategy to encourage representation update of $C_{cur}$ and $C_{old}$ at the same time. Thus our mask to $X_{cur}\cup X_{old}$ is $1_{C_{cur}\cup C_{old}}$. Similarly, we exclude other seen classes but not presented in the current mini-batch to avoid interference with their representations.
> >
> > The overall optimization objective of ER-ACE is
> > $$
> > L_{ce}(X_{bf}, Y_{bf} |C_{seen}) + L_{ce}(X_{cur}, Y_{cur} |C_{cur}),
> > $$
> > whereas ours is
> > $$
> > L_{ce}(X^{old} \cup X^{cur} , Y^{old} \cup Y^{cur} |C_{old} \cup C_{cur}),
> > $$
> >
> > where $L_{ce}(X, Y| C)$ is the cross-entropy loss with $\mathbb{1}_{C}$ as the binary mask on the data $X, Y$.
> >
> > We have added this analysis in our revised manuscript (see lines 353-355 and Appendix A.4.10).

---

> > > ### Comment · Reviewer_1W4v · 2024-11-19
> > >
> > > I would like to thank the authors for addressing my comments and providing clear clarifications. I now consider my concerns to be fully addressed. I believe this is a strong paper that deserves to be accepted.

---

> > > > ### Author Response · Authors · 2024-11-19
> > > > **Thank you**
> > > >
> > > > We thank the reviewer for finding our response satisfactory and are happy to know the very positive rating.

---

### Official Review · Reviewer_y3su · 2024-11-04

**Soundness:** 4
**Presentation:** 3
**Contribution:** 3
**Rating:** 8
**Confidence:** 4

**Summary:**

The paper presents MISA, an innovative approach for tackling General Continual Learning (GCL), arguably the most challenging form of Continual Learning. They do so by starting from a pre-trained model, refined with prompts, and introducing two key components:

ISA-FAM, which is applied at the start of each learning session (essentially, at the beginning of each GCL task). This component ensures proper prompt initialization. Here, "proper" means achieving both robust generalization capabilities and mitigating future forgetting. To enhance the learning effectiveness of prompts in this phase, the authors introduce a “prompt augmentation” technique, where prompts are trained using an approach involving an MLP that is later thrown away at the end of this phase..

Non-parametric Logit Masking: similar to prior CL methods, the authors propose a session mask that preserves the final activations of classes not present in the current learning session. Since GCL involves blurred task boundaries and often lacks task identifiers, they use a unique batch-level mask designed to function universally across scenarios.

The experimental validation utilizes widely recognized CL datasets, with a particularly rich ablation analysis that provides in-depth insights into the method’s effectiveness.

**Strengths:**

1) The method introduced by the authors meets their claim and proves to be the SOTA in GCL;
2) the ablation section is really rich and thorough, probing deeply every aspect of the method;
3) while the components used by the authors are not exactly novel per se, their combination and the proposed modifications prove to be interesting and effective;
4) by cleverly leveraging the pre-trained data, the authors eliminated the need for a replay buffer. At the same time, they showed that the model can benefit from storing past exemplars in case it is allowed;
5) the key components of MISA are plug and play, as shown by one of the ablations (Table 5), meaning that they could work out of the box if applied on other methods;
6) The code is available in the supplementary material, which greatly helps reproducibility.

**Weaknesses:**

1) While it is true that a buffer is not mandatory for MISA, it still depends on the pre-train data for the ISA-FAM phase. This still poses a limitation, as the pre-train data may not be available;
2) the experimental section lacks the case of a completely out-of-distribution dataset.

**Questions:**

1) In the ISA-FAM phase, the classification heads are trained with the prompts but later discarded (replaced by the previously trained heads) when the real learning session begins. Why is that? This way, I suppose prompts would "communicate incorrectly" with the classification heads, as the latter are replaced;
2) is there any reason why DualPrompt was selected as the prompting method? Did you consider more performing methods like Coda-Prompt[1] or HiDe-Prompt[2]?
3) you did cite [3] but not [4] for General Continual Learning, although the latter was published earlier. Is there any particular reason for this? Did you prefer the former over the latter in your GCL formulation?

[1] Smith, James Seale, et al. "Coda-prompt: Continual decomposed attention-based prompting for rehearsal-free continual learning." Proceedings of the IEEE/CVF Conference on Computer Vision and Pattern Recognition. 2023.

[2] Wang, Liyuan, et al. "Hierarchical decomposition of prompt-based continual learning: Rethinking obscured sub-optimality." Advances in Neural Information Processing Systems 36 (2024).

[3] Pietro Buzzega, Matteo Boschini, Angelo Porrello, Davide Abati, and Simone Calderara. Dark experience for general continual learning: a strong, simple baseline. 2020.

[4] Matthias De Lange, Rahaf Aljundi, Marc Masana, Sarah Parisot, Xu Jia, Ales Leonardis, Gregory Slabaugh, and Tinne Tuytelaars. A continual learning survey: Defying forgetting in classification tasks. PAMI, 44(7):3366–3385, 2021.

---

> ### Author Response · Authors · 2024-11-19
> **Authors' Response (Part I)**
>
> Thank you for the valuable comments. Below, we provide a point-to-point response to these comments and summarize the corresponding revisions. We would be happy to address any further questions.
>
> ### Q1: Dependence of pre-training data that may not be available.
>
> As shown in the response to Reviewer n9s5's Q3, the ISA-FAM phase requires only a small amount of data to construct the pseudo-CL tasks. We also considered the case without any pre-training data and replaced it with additional public datasets, where our approach still achieves consistent improvements over a range of strong baselines (see lines 228-232 and Appendix A.4.9).
>
> ### Q2: Experiment on a completely out-of-distribution (OOD) dataset.
>
> As shown in the response to Reviewer iGuV's Q2, we perform experiments in the medical image domain, which is an out-of-distribution of the dataset that has been used to pretrain the backbone, i.e., ImageNet-1k. Our approach achieves significant improvements. (See lines 512-514 and Table 6)
>
>
> ### Q3: Why is the classification head discarded?
>
> This is because the classes observed in the ISA-FAM phase and the actual continual learning sessions are semantically different. In other words, the classification head that projects the instructed representations to the class identity (also known as class prototypes) is no longer applicable as the class identity changes. We have further clarified this in the revised manuscript (see lines 226-228).
>
> ### Q4: Why DualPrompt was selected as the prompting method?
>
> In our original submission, we followed the implementation of the Si-Blurry paper [a], which employed DualPrompt as the representative baseline of prompt-based methods to develop their approach. When considering more advanced prompt-based methods, such as CODA-Prompt and HiDe-Prompt, we find that their specific designs are difficult to adapt to general continual learning (GCL), especially the challenge of blurry task boundaries.
>
> Specifically, HiDe-Prompt requires approximating and preserving class-wise distribution for subsequent representation recovery. This strategy is easy to implement in regular CL as all training samples for each class are provided together to calculate the class mean and covariance. This is however incompatible with GCL as only a portion of the training samples for each class are provided at a time. Other relevant methods of representation recovery, such as SLCA (ICCV 2023) and RanPAC (NeurIPS 2023), face similar problems.
>
> On the other hand, CODA-Prompt constructs expandable attention vectors, prompt keys, and prompt components for each task, where such parameters of old tasks are frozen when learning new tasks, coupled with orthogonality regularization between tasks to alleviate catastrophic forgetting. Although it could be implemented into the Si-Blurry setting, the effectiveness of freezing and orthogonality depends on the assumption of explicit task boundaries (i.e., inter-task classes are disjoint), making it sub-optimal with blurry task boundaries. We have empirically validated this claim, where CODA-Prompt only achieves 38.7\% on 5-task GCL with ImageNet-R and lags far behind our 50.89\%, and also behind the earlier work Dual-Prompt of 40.1\%. We include the additional CODA-Prompt GCL experiments in the response to reviewer n9s5's Q4, which supports our analysis by experimental evidence.
>
> Our analysis further suggests the importance of practical considerations when designing prompt-based methods for GCL, which constitutes our main motivation. We have added the above analysis in our revised manuscript (see lines 126-130).
>
>
> [a] Online class incremental learning on stochastic blurry task boundary via mask and visual prompt tuning. ICCV, 2023.

---

> > ### Author Response · Authors · 2024-11-19
> > **Authors' Response (Part II)**
> >
> > ### Q5: Citation of [3] and [4] for general continual learning (GCL).
> > We would respectfully point out that we have cited both [3] and [4] in our original submission (lines 40-42). Specifically, our work focuses on the particular challenges of online data streams and blurry task boundaries in the context of GCL. [4] is essentially a conceptual outline of the main characteristics underlying GCL, including general challenges of regular CL (e.g., *forward transfer*, *backward transfer*, *constant memory* and *no test time oracle*) and other specific ones (e.g., *online learning* and *no task boundaries*). [3] considers a particular GCL setting of blurry task boundaries, but fails to consider online data streams. In comparison, our work attempts to address all the above challenges within a broader context of GCL research.
> >
> > [3] Dark experience for general continual learning: A strong, simple baseline. NeurIPS, 2020.
> >
> > [4] A continual learning survey: Defying forgetting in classification tasks. PAMI, 2021.

---

> > > ### Author Response · Authors · 2024-11-24
> > > **Gentle reminder**
> > >
> > > Dear Reviewer y3su,
> > >
> > > If you have any further feedback or questions, please feel free to let us know. Your input is highly appreciated, and we look forward to hearing from you. Thank you once again for your time and consideration!

---

> > > ### Comment · Reviewer_y3su · 2024-11-24
> > >
> > > To the best of my knowledge, the setting proposed in [3] considers online data streams, as it allows just a single epoch on the proposed GCL scenario.
> > > Anyway, this was just a curiosity: I do not consider this crucial for the acceptance.

---

> > > > ### Author Response · Authors · 2024-11-24
> > > > **Thank you**
> > > >
> > > > We are grateful for your valuable comments and for your very positive rating.
> > > >
> > > > Moreover, thank you for pointing out the one-epoch experiments in [3]. We will be more careful when summarizing previous GCL works. We just want to share our understanding with you: in the main paper of [3], they only conducted the one-epoch experiment with the simple MNIST dataset, but not for more complex CIFAR-10 and Tiny-ImageNet datasets, which were experimented with 50 and 100 epochs, respectively (see section 4.1 of [3]). The additional one-epoch experiments for these two datasets are presented in Supplementary Materials F.3 of [3]. Interestingly, the authors had a rather *pessimistic* attitude toward the one-epoch protocol for complex datasets in GCL, as it leads to strong underfitting in training-from-scratch scenarios. This justifies the use of pretrained models in GCL to alleviate this underfitting and meet the guidelines of GCL in [4], as we illustrated in the Introduction section of our submission.

---

> > ### Comment · Reviewer_y3su · 2024-11-24
> >
> > Thank you for answering all my questions and concerns. I confirm my positive rating.

---

### Official Review · Reviewer_iGuV · 2024-11-10

**Soundness:** 3
**Presentation:** 3
**Contribution:** 1
**Rating:** 5
**Confidence:** 4

**Summary:**

This paper introduces MISA, a novel method that enhances prompt-based approaches in Generalized Continual Learning (GCL) by incorporating a forgetting-aware initial session adaptation and a non-parametric logit mask. Experiments demonstrate the proposed method's effectiveness and its ability to improve performance on three representative datasets: CIFAR-100, Tiny-ImageNet, and ImageNet-R.

**Strengths:**

1. The paper is well-structured and easy to understand.
2. The authors aimed to propose an effective methodology for a scenario that is closer to real-world conditions, compared to the common CL scenarios assumed by existing prompt-based CL methods.

**Weaknesses:**

1. The methodology appears to lack substantial novelty. The approach of using SAM for effective initialization and leveraging previously learned knowledge with masking for robustness against forgetting has been proposed and utilized numerous times in CL.
2. In the context of GCL, demonstrating the performance of a pre-trained model on natural images only with similar or closely related benchmarks weakens the paper's claims. It would be important to show that the SAM + masking with a pre-trained model approach remains strong when tested on benchmarks composed of datasets from significantly different domains, such as completely different views, 3D data, or medical imagery.
3. The scenario used in the paper is based on an already proposed (limited) GCL scenario that assumes blurry and task-free settings, which may not fully capture the (real) general scenario. I believe including experiments that reflect more realistic situations, such as not knowing the number of classes in the beginning or noisy datasets, would enhance the paper's contribution.

**Questions:**

Please address the concerns mentioned above in weaknesses.

---

> ### Author Response · Authors · 2024-11-19
> **Authors' Response (Part I)**
>
> Thank you for the valuable comments. Below, we provide a point-to-point response to these comments and summarize the corresponding revisions. We hope these responses provide sufficient reasons to raise the score. We would be happy to address any further questions.
>
> ### Q1: Novelty of the proposed approach.
>
> Our main motivation is to advance prompt-based methods in general continual learning (GCL), especially for online data streams and blurry task boundaries. Although prompt-based methods achieve excellent performance in regular CL settings, many critical issues (e.g., poor initial performance, limited generalizability, severe catastrophic forgetting, etc.) are clearly exposed under the requirements of GCL. In this regard, we demonstrate the importance of specialized designs of tunable parameters in prompt-based methods, i.e., the prompt parameters and the output layers.
>
> We respectfully remind the reviewer of our method design, i.e., Initial-session adaptation (ISA), forgetting-aware optimization (FAM), and non-parametric logit masking. To the best of our knowledge, we are the first to explore the benefits of prompt pre-training for CL. Moreover, we construct pseudo CL tasks to approximate forgetting-sensitive directions in flatness optimization, which stands for the key considerations of adapting flatness optimization in the context of CL. Therefore, our FAM differs from the standard SAM and is more adapted to GCL methods. We also re-design the learnable logit masking of state-of-the-art GCL baseline, making it more effective, easy to use, and avoid CL-relevant hyperparameters. As recognized by all the other reviewers, the proposed modifications are *interesting and useful* (Reviewer n9s5), *interesting and effective* (Reviewer y3su), and *noteworthy* (Reviewer 1W4v).
>
>
> ### Q2: Knowledge overlap between pre-training data and downstream tasks.
>
> We would respectfully point out that the impact of knowledge overlap between pre-training data and downstream tasks has been investigated in our original submission. Specifically, we consider a particular pre-trained checkpoint with 389 classes removed from ImageNet-1k that are similar to CIFAR-100 or Tiny-ImageNet. As shown in Table 6 of the main paper, our approach still achieves considerable improvements over all the baselines.
>
> As suggested, we further analyze our approach with more different downstream tasks such as medical images (i.e., the NCH  [a] dataset). We report the $A_{AUC}$ scores over runs with 5 random seeds in the following table. We observe substantial improvement over existing methods. In particular, in the more challenging 5-task setting, our method not only surpasses existing work by a large margin, but also has a much smaller standard deviation. This showcases the robustness of our proposed method in out-of-distribution domain. We included two more out-of-distribution datasets for autonomous driving (i.e., the GTSRB [b] dataset) and fine-art style classification (i.e., the WikiArt [c] dataset) to further validate the performance of our method in unseen or unrelated domains.
>
> | Method     | NCH (5-Task)  | NCH (3-Task)  | GTSRB (5-Task)     | WikiArt style (5-Task) |
> | ---------- |:------------- |:------------- |:--------- |:------- |
> | EWC        | 52.0+-12.4    | 74.0+-4.0     | 52.8+-8.1 | 31.4+-4.6    |
> | DualPrompt | 60.6+-11.7    | 72.6+-3.3     | 42.7+-7.3 | 32.9+-3.8    |
> | MVP        | 64.0+-10.1    | 76.4+-2.1     | 51.1+-4.2 | 33.0+-4.0    |
> | Ours       | **71.2+-3.3** | **81.3+-5.7** | **62.9+-8.3** | **40.0+-4.4**    |
>
> We have added these results in our revised manuscript (see lines 512-514 and Table 6).
>
> [a] Jakob Nikolas Kather, Niels Halama, and Alexander Marx. 100,000 histological images of human colorectal cancer and healthy tissue, May 2018.
>
> [b] Stallkamp, Johannes, et al. "The German traffic sign recognition benchmark: a multi-class classification competition." The 2011 international joint conference on neural networks. IEEE, 2011.
>
> [c] Saleh, Babak, and Ahmed Elgammal. "Large-scale classification of fine-art paintings: Learning the right metric on the right feature." arXiv preprint arXiv:1505.00855 (2015).

---

> > ### Author Response · Authors · 2024-11-19
> > **Authors' Response (Part II)**
> >
> > ### Q3: More realistic situations such as not knowing the number of classes in the beginning or noisy datasets.
> >
> > In fact, the problem setup considered in this work does **not** know the number of classes in advance, which means that each task may include an arbitrary number of new and old classes in an arbitrary ratio. In experiments, we randomly construct multiple such task sequences and report the average performance. Again, all methods are not allowed to use the information of unseen class in continual learning.
> >
> > Besides, as discussed in the response to Reviewer n9s5's Q1, we also evaluate continual learning with one class at a time, and our approach achieves significant improvements. Note that any class-incremental scenario can be decomposed to one-class increments, without assuming the number of classes in each new task. This challenging setting demands the model to swiftly adapt and learn new knowledge about a new single class without forgetting previous knowledge that is underrepresented in the current data stream. We have added more details to clarify it in our revised manuscript (see lines 365-366 and Appendix A.4.8).

---

> > > ### Author Response · Authors · 2024-11-23
> > > **Two additional out-of-distribution datasets for Q2**
> > >
> > > We have finished the experiments of two additional out-of-distribution datasets for autonomous driving (i.e., the GTSRB [b] dataset) and fine-art painting style classification (i.e., the WikiArt [c] dataset), and added them in the response to Q2.
> > >
> > > [b] Stallkamp, Johannes, et al. "The German traffic sign recognition benchmark: a multi-class classification competition." The 2011 international joint conference on neural networks. IEEE, 2011.
> > >
> > > [c] Saleh, Babak, and Ahmed Elgammal. "Large-scale classification of fine-art paintings: Learning the right metric on the right feature." arXiv preprint arXiv:1505.00855 (2015).

---

> > > > ### Author Response · Authors · 2024-11-24
> > > > **Gentle reminder**
> > > >
> > > > Dear Reviewer iGuV,
> > > >
> > > > If you have any further feedback or questions, please feel free to let us know. Your input is highly appreciated, and we look forward to hearing from you. Thank you once again for your time and consideration!

---

### Official Review · Reviewer_n9s5 · 2024-11-13

**Soundness:** 3
**Presentation:** 3
**Contribution:** 3
**Rating:** 5
**Confidence:** 4

**Summary:**

In this paper, the authors proposed a new prompt based method for general continual learning. Here they utilize forgetting-aware initial session adaption that employs pretraining data to initialize prompt parameters to improve generalizability. Additionally, they propose to use a simple non-parametric logit mask at the output layers to mitigate catastrophic forgetting. With comprehensive experiments and analysis they show the effectiveness of the method in GCL setup.

**Strengths:**

1. The idea of the paper is clearly presented with comprehensive experiments, ablations, and analysis.
2. Though the paper uses concepts from other previous works the combination of the ideas and their effective implementation in the GCL setup is interesting and useful for the continual learning community.

**Weaknesses:**

1. The paper aims to address the challenging general continual learning (GCL) setup. However, in this setup, the classes can appear one by one and the model may need to learn one class at a time. No discussion on how to handle such a situation is presented in the paper.
2. Though the authors presented experiment with blurry task/class boundaries, in true GCL setup the labels of the classes might not be known ahead of time. In such cases, the model needs to detect novel classes and then learn that new classes (see [1]). How the current algorithm would handle such scenarios is not discussed in the paper.
3. The algorithm assumes the availability of the pertaining data. This is a strict constraint. In most practical cases this would not be available. For instance for large opensourced pre-trained vision, vision-language, and language models, the corresponding pretraining data is not publicly released. In the absence of any pertaining data or with very limited accessibility of pertaining data how this algorithm will perform?
4. The paper failed to cite and/or compare with many related works. For example: [2,3,4].


[1] M. Gummadi, D. Kent, J. Mendez, and E. Eaton, “SHELS: Exclusive
Feature Sets for Novelty Detection and Continual Learning Without Class
Boundaries,” PMLR Conference on Lifelong Learning Agents, pp. 1065-
1085, 2022
[2] Coda-prompt: Continual decomposed attention-based prompting for rehearsal-free continual learning. In CVPR, pp. 11909–
11919, 2023.
[3] RanPAC: Random Projections and Pre-trained Models for Continual Learning. (https://arxiv.org/abs/2307.02251)
[4] Visual Prompt Tuning in Null Space for Continual Learning (https://arxiv.org/abs/2406.05658)

**Questions:**

See weakness.

---

> ### Author Response · Authors · 2024-11-19
> **Authors' Response (Part I)**
>
> Thank you for the valuable comments. Below, we provide a point-to-point response to these comments and summarize the corresponding revisions. We hope these responses provide sufficient reasons to raise the score. We would be happy to address any further questions.
>
> ### Q1: Continual learning of one class at a time.
>
> Following this suggestion, we now evaluated general continual learning with one class at a time. Specifically, we conducted 100-task CIFAR-100 and 200-task ImageNet-R experiments with 5 random seeds and report the $A_{AUC}$ score in the following table. Note that the tasks are not strictly disjoint, as the data stream can still contain data from previous tasks due to GCL's blurry task boundary requirements. This challenging setting demands the model to swiftly adapt and learn new knowledge about a new single class without forgetting previous knowledge that is underrepresented in the current data stream. For instance, the performance of EWC and MVP drops dramatically to less than 5\% in this challenging setting with ImageNet-R. As shown in the table, our approach achieves consistent improvements over all the baselines. We have added it in our revised manuscript (see lines 365-366 and Appendix A.4.8).
>
> | Method     | ImageNet-R    | CIFAR-100     |
> | ---------- |:------------- |:------------- |
> | EWC        | 2.3+-1.1      | 68.1+-0.4     |
> | DualPrompt | 25.6+-5.6     | 80.4+-0.4     |
> | MVP        | 4.0+-1.4      | 78.7+-0.1     |
> | Ours       | **31.1+-0.6** | **85.3+-0.1** |
>
> ### Q2：To detect unknown classes and then learn that classes with annotation.
>
> As analyzed in the paper mentioned [1] and in two pioneering works [a,b], this setting is formally known as the open-world problem, which corresponds to a combination of (1) novel class discovery and (2) continual learning. These two aspects are essentially plug-and-play and complementary to each other, since continual learning requires human annotation of unknown classes after they have been discovered. Therefore, we respectfully argue that novel class discovery is out of scope of our focus (i.e., advancing prompt-based methods in continual learning with online data streams and blurry task boundaries) and may serve as a plug-in module of our approach. Interestingly, an extended version [c] of HiDe-Prompt (NeurIPS 2023) has explored the combination of OOD detection and prompt-based continual learning, which validates this vision. We believe this combination remains a promising direction for our future research and have added a discussion in our revised manuscript (see lines 142-144).
>
> [1] SHELS: Exclusive feature sets for novelty detection and continual learning without class boundaries. CoLLAs, 2022.
>
> [a] Towards open world object detection. CVPR, 2021.
>
> [b] Learngene: From open-world to your learning task. AAAI, 2022.
>
> [c] HiDe-PET: Continual learning via hierarchical decomposition of parameter-efficient tuning. arXiv, 2024.

---

> > ### Author Response · Authors · 2024-11-19
> > **Authors' Response (Part II)**
> >
> > ### Q3: Availability of pre-training data.
> >
> > In our preliminary experiments, we find that although using full pretraining dataset yields the best performance, using only a small fraction of pre-training data (e.g., 100 classes out of the entire ImageNet-1k) to initialize prompts also achieved desirable performance. This is because the objective of our forgetting-aware initial session adaption is to initialize prompts with general and transferable knowledge, not necessarily as strong as that of the PTM but significantly better than the random initialization.
> >
> > As suggested, we further evaluated our method in cases where pretraining data is partially or even completely unavailable in the following table. Existing works (i.e., EWC, DualPrompt, MVP) only depend on the PTM whereas ours is also impacted by the ISA. We report the $A_{AUC}$ score from 5-task GCL on ImageNet-R with 5 random seeds.
> >
> > The first column corresponds to performing ISA with ImageNet-100 (IN100) with the backbone pretrained on ImageNet-1k (IN1k). We notice that with only 10\% of the pretraining dataset, our method can obtain comparable performance with that using full IN-1k: 49.5\% v.s. 50.8\%. The second column shows ISA on IN100 with PTM pretrained in a CLIP style ([Model card](https://huggingface.co/timm/vit_base_patch16_clip_224.openai)). The pretraining data contains YFCC100M and web-crawled data, which is an ideal example to simulate the closed-source models whose training data is completely unavailable. We can still achieve a substantial performance gain when we perform ISA on a publicly available and semantically unrelated dataset: IN100. This justifies the flexibility of our proposed method.
> >
> > We note that a more diverse and generic dataset helps ISA to obtain more generalizable prompt parameters. As shown in the third column, when we perform ISA on a fine-grained dataset (CUB200), the performance drops significantly, i.e. 43.4\% v.s. 49.5\%.
> >
> > We have added the additional experiments and analysis in our revised manuscript (see lines 228-232 and Appendix A.4.9).
> >
> >
> > | Method     | PTM(IN1k)+ISA(IN100) | PTM(YFCC100M) + ISA(IN100) | PTM(IN1k)+ISA(CUB200) |
> > |:---------- |:-------------------- |:-------------------------- |:--------------------- |
> > | EWC        | 31.5+-1.0            | 30.6+-1.3                  | 31.5+-1.0             |
> > | DualPrompt | 40.1+-1.2            | 37.9+-1.0                  | 40.1+-1.2             |
> > | MVP        | 40.6+-1.2            | 34.0+-1.3                  | 40.6+-1.2             |
> > | Ours       | **49.5+-1.3**          | **49.3+-1.2**              | **43.4+-1.9**         |

---

> > > ### Author Response · Authors · 2024-11-19
> > > **Authors' Response (Part III)**
> > >
> > > ### Q4: Cite and/or compare with more related works.
> > >
> > > Thank you for providing these related works. As discussed in the response to Reviewer y3su's Q4, existing prompt-based methods focus on general challenges of regular CL, and remain sub-optimal in addressing particular challenges of GCL.
> > >
> > > Specifically, CODA-Prompt [2] constructs expandable attention vectors, prompt keys and prompt components for each task, where such parameters of old tasks are frozen when learning new tasks, coupled with orthogonality regularization between tasks to alleviate catastrophic forgetting. The effectiveness of freezing and orthogonality depends on the assumption of explicit task boundaries (i.e., inter-task classes are disjoint), creating undesirable results with blurry task boundaries. We have empirically validated this claim, where CODA-Prompt only achieves 38.7% on ImageNet-R and lags far behind our approach (50.89%), and also Dual-Prompt. Similarly, NSP$^2$ [4] performs prompt gradient orthogonal projection to ensure no interference between tasks, which is also inconsistent with blurry task boundaries.
> > >
> > > On the other hand, RanPAC [3] requires approximating and preserving class-wise distributions for subsequent representation recovery. This strategy is easy to implement in regular CL as all training samples for each class are provided together to calculate the class mean and covariance, whereas is incompatible with GCL as only a portion of the training samples for each class are provided at a time. Other relevant methods of representation recovery such as SLCA (ICCV 2023) and HiDe-Prompt (NeurIPS 2023) face similar problems.
> > >
> > > In particular, we have implemented CODA-Prompt in 5-task GCL and presented the results ($A_{AUC}$) as follows:
> > >
> > > | Method                                   | ImageNet-R |
> > > |:---------------------------------------- |:---------- |
> > > | Dual-Prompt                              | 40.1+-4.2  |
> > > | CODA-Prompt                              | 38.7+-2.3  |
> > > | CODA-Prompt /w orthogonal regularization | 30.5+-1.1  |
> > > | CODA-Prompt + Ours                       | **48.4+-2.0**  |
> > >
> > > Note that in the latest implementation of CODA-Prompt, the orthogonality regularization is omitted. To validate our above analysis, we further conduct experiments with the original orthogonal regularization as illustrated in the paper. We observe a drop of performance as expected, which confirms that the blurry task boundaries violates the orthogonality assumption. Moreover, we conduct ISA on the attention, key and prompt parameters of CODA-Prompt on ImageNet-100 and apply our logit masking to CODA-Prompt. We observe a boost of performance, denoted as *CODA-Prompt + Ours* in the table. This highlights the plug-in nature of our proposed method.
> > >
> > > We have added the above analysis in our revised manuscript (see lines 126-130).
> > >
> > >
> > > [2] CODA-Prompt: Continual decomposed attention-based prompting for rehearsal-free continual learning. CVPR, 2023.
> > >
> > > [3] RanPAC: Random projections and pre-trained models for continual learning. NeurIPS, 2023.
> > >
> > > [4] Visual Prompt Tuning in Null Space for Continual Learning. NeurIPS, 2024.

---

> > > > ### Author Response · Authors · 2024-11-24
> > > > **Gentle reminder**
> > > >
> > > > Dear Reviewer n9s5,
> > > >
> > > > If you have any further feedback or questions, please feel free to let us know. Your input is highly appreciated, and we look forward to hearing from you. Thank you once again for your time and consideration!

---

### Author Response · Authors · 2024-11-20
**Overall response**

We sincerely appreciate the valuable feedback provided by all the reviewers. It is encouraging to see many positive comments, especially noting that our presentation is *clear* (Reviewers n9s5, iGuV, 1W4v), our method is *novel* (Reviewers n9s5, y3su, 1W4v) and *effective* (Reviewers n9s5, iGuV, y3su, 1W4v), our experiments are *comprehensive* (Reviewers n9s5, y3su, 1W4v), etc.

We have provided a point-to-point response to all the comments, summarized the corresponding changes, and uploaded a revised version of our manuscript. Our responses and revisions are summarized below.

(1) We have analyzed a variety of advanced prompt-based methods tailored for regular CL, showing their shortcomings for GCL in terms of representation recovery and orthogonality regularization.

(2) We have added more explanations and experiments to address the scenarios where pretraining data is partially or even completely unavailable.

(3) We have included additional explanations and experiments to address scenarios where the pretraining data and subsequent CL tasks are significantly different, as well as the challenging one-class increments.

(4) We have further clarified our framework to address the challenges of CGL, highlighting its innovation compared to previous efforts in regular CL.

We hope these responses provide sufficient materials to address the reviewers' concerns. We would be happy to answer any further questions.

---

### Author Response · Authors · 2024-12-04
**Final overall response and Gratitude to the reviewers**

Dear Reviewers,

Thank you for your thoughtful feedback and efforts during the review process of our submission. We would like to kindly address two points as the rebuttal phase concludes.

For Reviewer n9s5 and iGuV: Since no further questions were raised during the rebuttal discussion, we hope this indicates that we have satisfactorily addressed your initial remarks. We would be grateful if you could consider re-evaluating our work and potentially raising the overall score.

For Reviewer y3su and 1W4v: Your engagement during the rebuttal has been invaluable in ensuring that our work receives a thorough and fair evaluation.

Once again, we are very grateful for your time, thoughtful reviews, and dedication to ensuring a constructive and fair evaluation process.

Best regards,

Authors of paper ID 7052

---

### Meta-Review · Area_Chair_AH2z · 2024-12-19

**Metareview:**

The paper investigates general continual learning and introduces MISA (Mask and Initial Session Adaptation) to advance prompt-based methods. Strong empirical performance is achieved due to a combination of two strategies, refining prompt parameter initialization and output layer masking. The paper has several strengths, including the plug-and-play nature of the approach, its strong empirical performance, a thorough ablation study, clear presentation, and interesting combination of methodology. The weaknesses and points for improvement revolved around additional comparisons, e.g. on completely out-of-distribution scenarios and in scenarios were pre-training data is not available, alongside concerns on the training set-up. These points were addressed in the discussion phase and additional experiments added for clarification.

The AC agrees with the reviewers that there are several strong ideas in the paper and that the paper provides a meaningful empirical improvement in a very challenging to solve scenario. The AC votes to accept the paper.

**Additional Comments On Reviewer Discussion:**

The authors have provided extensive responses, clarifications and requested additions throughout the discussion period. Two of the reviewers have acknowledged these efforts and were satisfied to retain their acceptance recommendations. They consider the paper to have a strong contribution and the discussion phase to have further improved it.

The other two reviewers have not engaged in the discussion phase. The AC chooses to weigh the respective borderline scores less heavily as
* reviewer iGuV seems to have had a misunderstanding on the investigated scenario and has in consequence conjectured the novelty to be severely limited. These were two out of the three main concerns of the initial review, which are not in line with the other reviewers’ assessments. The authors have clarified these in the rebuttal and have included experimental additions to address the third raised points. However, the reviewer has not acknowledged the rebuttal. The AC believes the points have been sufficiently clarified.
* reviewer n9s5 has requested the addition of another experiment in a modified scenario, as well as a clarification on the role of pre-trained data. The authors have provided the requested additional results. The authors have also included further related work comparisons. All shown results seem to clearly demonstrate the paper’s benefits and the AC believes the reviewer’s points to have been addressed sufficiently. The reviewer unfortunately did not acknowledge the rebuttal. In addition, the reviewer seems to have lowered their initial score of 6 to a score of 5 without any stated reason after posting the initial review. This choice is unclear to the AC.

Following the above reasons the AC choses to accept the paper.

---

### Decision · Program_Chairs · 2025-01-22

Accept (Poster)